# Recruitment of two Ndc80 complexes via the CENP-T pathway is sufficient for kinetochore functions

Yusuke Takenoshita [1], Masatoshi Hara [1] & Tatsuo Fukagawa [1✉]

To form functional kinetochores, CENP-C and CENP-T independently recruit the KMN (Knl1C, Mis12C, and Ndc80C) network onto the kinetochores. To clarify the functions of the KMN network on CENP-T, we evaluated its roles in chicken DT40 cell lines lacking the CENP-C-KMN network interaction. By analyzing mutants lacking both CENP-T-Mis12C and CENP-C-Mis12C interactions, we demonstrated that Knl1C and Mis12C (KM) play critical roles in the cohesion of sister chromatids or the recruitment of spindle checkpoint proteins onto kinetochores. Two copies of Ndc80C (N-N) exist on CENP-T via Mis12C or direct binding. Analyses of cells specifically lacking the Mis12C-Ndc80C interaction revealed that N-N is needed for proper kinetochore-microtubule interactions. However, using artificial engineering to directly bind the two copies of Ndc80C to CENP-T, we demonstrated that N-N functions without direct Mis12C binding to Ndc80C in native kinetochores. This study demonstrated the mechanisms by which complicated networks play roles in native kinetochores.

[1] Graduate School of Frontier Biosciences, Osaka University, Suita, Osaka 565-0871, Japan.  ✉email: tfukagawa@fbs.osaka-u.ac.jp

Chromosomes replicated during the S-phase must be divided into daughter cells during mitosis to transmit the genetic information to the progeny. In eukaryotes, this process is called chromosome segregation, which is achieved by the attachment of sister chromatids to the bipolar mitotic spindles. The spindle microtubules bind to a large protein complex called kinetochore, which is formed on the centromere of each sister chromatid, to ensure the accurate segregation of the chromosomes[1–4].

Kinetochores are composed of two major complexes. One complex, which consists of 16 protein components, forms a base on the centromeric chromatin for the formation of kinetochores and is called the constitutive centromere-associated network (CCAN), which constitutively localizes to the centromere throughout the cell cycle[2,3,5,6]. From the late $G_2$ phase to mitosis, another large complex starts to associate with CCAN to form a fully functional kinetochore. This complex contains the Knl1 complex (Knl1C), Mis12 complex (Mis12C), and Ndc80 complex (Ndc80C), which form the KMN network[2,6]. In the KMN network, Ndc80C directly associates with spindle microtubules[7–9], Knl1C appears to make an adapter for various proteins, including spindle checkpoint proteins[10–12], and Mis12C recruits Ndc80C and Knl1C[2,6]. The linkage between CCAN and KMN bridges the centromeric chromatin and spindle microtubules to facilitate accurate chromosome segregation.

CCAN is divided into several subgroups[13–16], including centromere protein (CENP)-C[17–22], CENP-T-W-S-X[5,23,24], CENP-L-N[25–29], CENP-H-I-K-M[14,30,31], and CENP-O-P-Q-R-U[16,32] complexes. Among the CCAN proteins, CENP-C and the CENP-T-W-S-X complex bind to both centromeric chromatin and the KMN network[1,33–35]. Based on these studies and the results of artificial tethering of CENP-C or CENP-T into a noncentromeric region, two parallel pathways for the recruitment of the KMN network onto the kinetochores in vertebrate cells have been proposed: the CENP-C and CENP-T pathways[1,3,4,34–37].

The kinetochore is a multi-protein complex with complicated protein-protein networks, which often exhibit redundant functions. Therefore, it is essential to dissect the functional roles of each complex in the native kinetochores. As some proteins contribute to multiple functions via interaction with different partners, the simple knockout approach for a target protein may cause misinterpretation of its protein function owing to the indirect effects of the knockout. Therefore, to evaluate the precise role of each complex or protein in native kinetochores, we must characterize a narrow functional domain in each protein and remove the redundancies to assess the function of an individual protein (complex).

We previously demonstrated that CENP-C-KMN network binding is dispensable for cell viability and kinetochore assembly in chicken DT40 cells[35]. In contrast, the CENP-T-KMN network interaction is essential for cell viability, suggesting that the CENP-T pathway is a major pathway in chicken cells[35–38]. However, the mechanism by which the KMN network coordinates its functions on the CENP-T pathway remains unclear. Particularly, while Mis12C binds to Knl1C and Ndc80C, it is unknown whether these interactions in the KMN network are essential for each function. Furthermore, in addition to Knl1C-Mis12C-Ndc80C (KMN) formation, an additional copy of Ndc80C directly binds to CENP-T[36,37,39,40], and one KMN unit and one Ndc80C forms KMN-N in the network on CENP-T[6,41]. The KMN network appears to play multiple roles, including in the binding of microtubules, spindle checkpoint functions, and cohesion of sister chromatids. However, it is unclear whether each function is coordinated by the entire network or if the network can be separated into subcomplexes to perform individual functions. Moreover, even if the entire network is separable, the specific combinations of components responsible for each function remain ambiguous. Therefore, it is critical to clarify these questions to understand kinetochore functions.

In this work, to address these questions, we examined the different roles of the KMN network on the CENP-T pathway in cells lacking the CENP-C-KMN network interaction, which exhibits some redundant functions to CENP-T. We constructed cell lines lacking the CENP-T-Mis12C interaction in the absence of the CENP-C-KMN network interaction and demonstrated defects in sister chromatid cohesion, reduction in the levels of Ndc80C, or loss of Knl1C and Bub1 in these cells. These results suggest that Mis12C-Knl1C (KM) interaction on the CENP-T pathway is critical for the recruitment of checkpoint proteins and Ndc80C or the cohesion of sister chromatids. While KM on CENP-T recruits Ndc80C, CENP-T directly binds to additional Ndc80C. We then constructed cell lines that specifically lacked the Ndc80C-binding site in Mis12C and demonstrated that one copy of Ndc80C on CENP-T was insufficient, suggesting that two copies of Ndc80C (N-N) are necessary for the functional kinetochore-microtubule interaction in DT40 cells. This suggests that the Mis12C-Ndc80C interaction is necessary for the microtubule-binding function of Ndc80C. However, using artificial engineering to directly bind the two copies of Ndc80C to CENP-T, we demonstrated that Ndc80C binding to Mis12C is not essential for microtubule binding. Our data indicate that the KMN network usually forms KMN-N (one copy of Ndc80C binds to KM as a KMN unit and the other directly binds CENP-T) on CENP-T for its functions, but KM-N-N (two copies of Ndc80C directly bind to CENP-T and KM independently binds to CENP-T) is functional in the network. We also demonstrated that Ndc80C binding to Mis12C is not essential for microtubule binding in human cells. Since the Ndc80C-Mis12C interaction is thought to be critical for the kinetochore functions[7,42,43], our observations provide new insights for kinetochore studies and explain how the KMN network plays an essential role in kinetochore functions via the CENP-T pathway.

## Results

**The CENP-T-Mis12C interaction is essential on the CENP-T pathway.** Mis12C binds to both CENP-C and CENP-T, and we have previously shown that the CENP-T-Mis12C interaction is much more critical than the CENP-C-Mis12C interaction in chicken DT40 cells[35]. However, deletion of the Mis12C-binding region (amino acids: aa 121-240 region) in chicken CENP-T did not lead to complete lethality in DT40 cells, although it caused a growth delay. This might be because Mis12C on CENP-C has redundant roles in kinetochore functions. We initially generated a cell line in which Mis12C-binding regions in both CENP-C and CENP-T were deleted (Fig. 1a-d). Since the Mis12C-binding site (N-terminal 73 aa region) in chicken CENP-C is dispensable in DT40 cells, we introduced mScarlet-fused CENP-C$^{\Delta 73}$ into one endogenous $\beta$-actin allele with disrupting the endogenous CENP-C gene using CRISPR/Cas9-mediated genome editing (Supplementary Fig. 1a-e). We chose the $\beta$-actin locus to obtain stable expression of mScarlet-fused CENP-C$^{\Delta 73}$, since the $\beta$-actin is ubiquitously expressed. Using this cell line, we introduced a cDNA for 3X FLAG-tagged CENP-T lacking the Mis12C-binding region (CENP-T$^{\Delta 121-240}$) into one CENP-T allele (Supplementary Fig. 1f, g), and another allele was replaced with a drug resistance gene, and a wild-type (WT) CENP-T transgene was expressed under the control of a tetracycline (Tet)-responsive promoter (Tet-CENP-T, Fig. 1d). We called this line conditional knockout (cKO)-CT (Tet-Off) -CC$^{\Delta 73}$/CENP-T$^{\Delta 121-240}$, and CENP-T and CENP-C were completely replaced with 3X FLAG-CENP-T$^{\Delta 121-240}$ and mScarlet-CENP-C$^{\Delta 73}$, respectively, after Tet addition in this

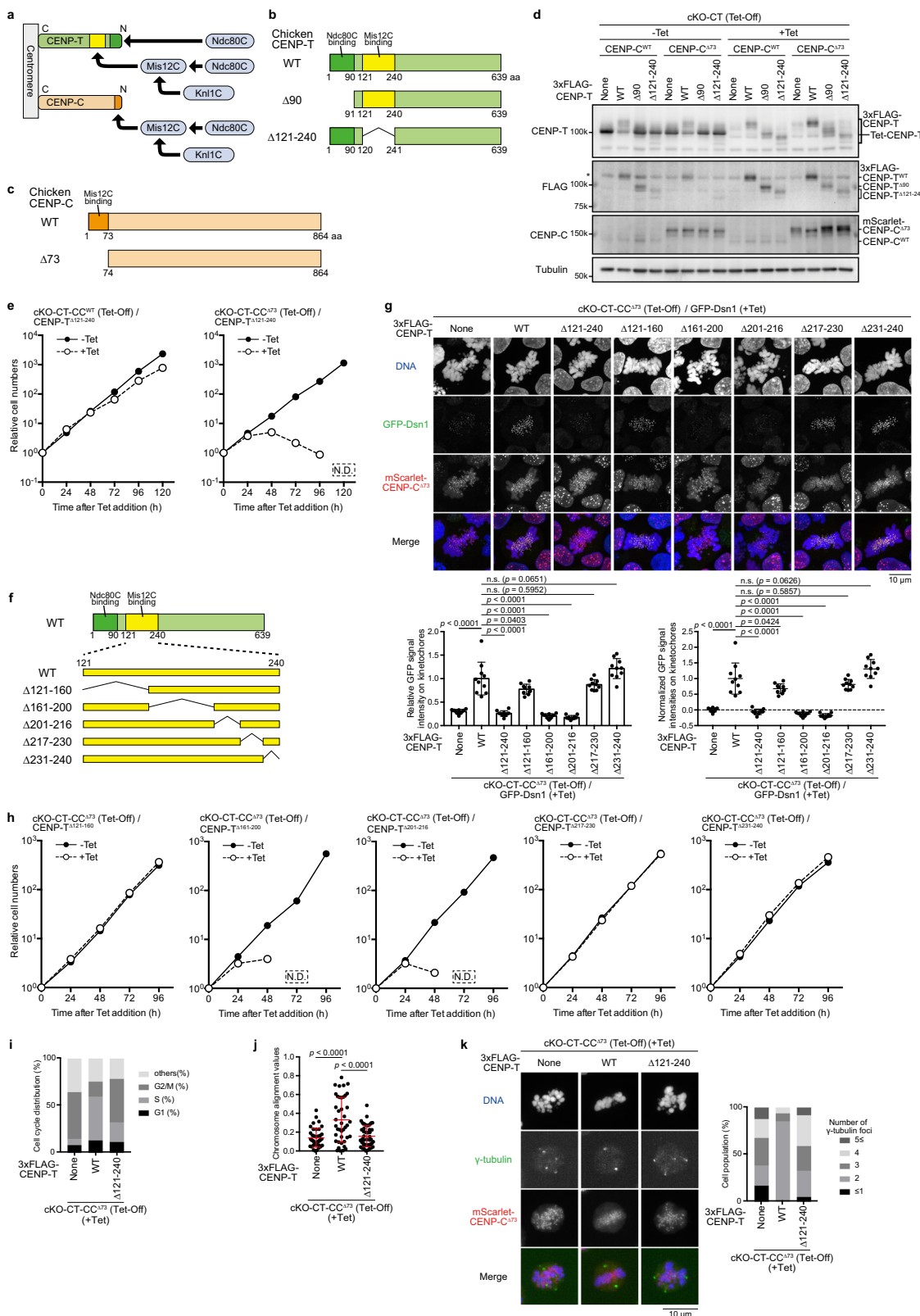

cell line (Fig. 1d). Using different combinations of CENP-T and CENP-C, we also generated various cell lines, such as cKO-CT-CC$^{\Delta 73}$/CENP-T$^{\Delta 90}$ or cKO-CT-CC$^{WT}$/CENP-T$^{\Delta 121-240}$ cells (Supplementary Fig. 1h). All cell lines generated in this study are summarized in Supplementary Table 1. We note that the CENP-T N-terminal region (aa 1-90) contains an Ndc80C-binding region (Fig. 1b), and CENP-T$^{\Delta 90}$ does not directly bind to Ndc80C[35,39].

After confirming the expression of each transgene using immunoblot analysis (Fig. 1d), we analyzed the growth of each cell line in the absence or presence of Tet. In the absence of Tet (expression of Tet-CENP-T), all cell lines grew well. After Tet addition, cKO-CT-CC$^{WT}$/CENP-T$^{\Delta 90}$ cells died by ~96 h (Supplementary Fig. 1i). However, cKO-CT-CC$^{WT}$/CENP-T$^{\Delta 121-240}$ cells were still viable in the presence of Tet (Fig. 1e), although

**Fig. 1 Disruption of the CENP-T-Mis12C interaction in cells lacking the CENP-C-Mis12C interaction leads to severe growth defects. a** Recruiting pathway of KMN network via CENP-C or CENP-T. **b** Schematic representation of chicken CENP-T. Ndc80C- and Mis12C-binding domains were deleted in CENP-T$^{\Delta 90}$ and CENP-T$^{\Delta 121-240}$, respectively. **c** Schematic representation of chicken CENP-C. The Mis12C-binding domain was deleted in CENP-C$^{\Delta 73}$. **d** Immunoblot analyses of various CENP-C or CENP-T in cKO-CENP-T (Tet-Off) cells. Expression of the CENP-T transgene (Tet-CENP-T) was terminated by 24 h after Tet addition. mScarlet-CENP-C$^{\Delta 73}$ was expressed in CENP-C knockout cells. The 3X FLAG-fused CENP-T mutants were stably expressed. CENP-C migration was slightly slow, due to CENP-C phosphorylation in mitosis. Alpha-tubulin (tubulin) was used as the loading control. The asterisks indicate nonspecific bands. None depicts parental cKO-CENP-T cells. **e** The growth of cKO-CENP-T cells expressing 3X FLAG-CENP-T$^{\Delta 121-240}$ in the absence or presence of Tet (−Tet or +Tet) with CENP-C$^{WT}$ (left) or mScarlet-CENP-C$^{\Delta 73}$ (right). **f** Schematic representation of each CENP-T mutant. **g** GFP-Dsn1 levels at kinetochores in cKO-CENP-T cells expressing mScarlet-CENP-C$^{\Delta 73}$ and various CENP-T mutants in the presence of Tet for 30 h. None indicates parental cKO-CENP-T/CENP-C$^{\Delta 73}$/GFP-Dsn1 cells. Subtract values are presented using signal intensities of None. DNA was stained with DAPI. Error bars show the mean and standard deviation. *p* values were calculated by one-way ANOVA followed by Tukey's test. **h** The growth of cKO-CENP-T cells expressing CENP-C$^{\Delta 73}$ and either CENP-T$^{\Delta 121-160}$, CENP-T$^{\Delta 161-200}$, CENP-T$^{\Delta 201-216}$, CENP-T$^{\Delta 217-230}$, or CENP-T$^{\Delta 231-240}$ in the absence or presence of Tet. Some time points (N.D.) were missing because of no viable cells. **i** Cell cycle distribution of cKO-CENP-T/CENP-C$^{\Delta 73}$ (None), cKO-CENP-T/CENP-C$^{\Delta 73}$ expressing CENP-T$^{WT}$ or CENP-T$^{\Delta 121-240}$ in the presence of Tet for 30 h. **j** Chromosome alignment values in cKO-CENP-T/CENP-C$^{\Delta 73}$ (None), cKO-CENP-T/CENP-C$^{\Delta 73}$ expressing CENP-T$^{WT}$, or CENP-T$^{\Delta 121-240}$. The cells were cultured in the presence of Tet for 30 h and MG132 for last 3 h. Error bars indicate the mean and standard deviation. *p* values were calculated as in **g**. **k** Numbers of spindle pole stained by anti-γ-tubulin in cKO-CENP-T/CENP-C$^{\Delta 73}$ (None), cKO-CENP-T/CENP-C$^{\Delta 73}$ expressing CENP-T$^{WT}$, or CENP-T$^{\Delta 121-240}$ in the presence of Tet for 30 h.

they showed a growth delay, which is consistent with the results of our previous study[35]. We further tested the growth of cKO-CT-CC$^{\Delta 73}$/CENP-T$^{\Delta 121-240}$ cells in the presence of Tet and found that these cells completely died by ~120 h after Tet addition (Fig. 1e). These results indicate that Mis12C on CENP-C has redundant roles in the kinetochore functions, although it is not sufficient to completely compensate for the lack of CENP-T-Mis12C interaction. Therefore, to precisely evaluate the Mis12C function on the CENP-T pathway in native kinetochores, we must analyze it in the absence of CENP-C-KMN network interaction (expression of CENP-C$^{\Delta 73}$ instead of CENP-C$^{WT}$).

Although we defined the aa 121-240 region as the Mis12C-binding region of chicken CENP-T, we attempted to define a more precise region of CENP-T required for Mis12C binding, using cKO-CT-CC$^{\Delta 73}$ cells (Fig. 1f). In addition to centromeric signals, some mScarlet-fused CENP-C$^{\Delta 73}$ localized to the non-centromeric chromosome region in these cells (Fig. 1g). This may be due to overexpression of mScarlet-fused CENP-C$^{\Delta 73}$ under the control of *β*-actin promoter. Since we confirmed that cells expressing mScarlet-fused CENP-C$^{\Delta 73}$ are viable, mScarlet-fused CENP-C$^{\Delta 73}$ should be functional. We also introduced GFP-Dsn1 (a component of Mis12C) into endogenous *Dsn1* alleles in these cells (Supplementary Fig. 1j-l). We note that cKO-CT-CC$^{\Delta 73}$ cells (None in Fig. 1g) died before the complete depletion of CENP-T. This explains why GFP-Dsn1 is still visible in cKO-CT-CC$^{\Delta 73}$ cells, and we also presented normalization data for GFP-Dsn1 based on data from cKO-CT-CC$^{\Delta 73}$ cells. As shown in Fig. 1g, GFP-Dsn1 levels were significantly reduced in either cKO-CT-CC$^{\Delta 73}$/CENP-T$^{\Delta 161-200}$ or cKO-CT-CC$^{\Delta 73}$/CENP-T$^{\Delta 201-216}$ cells, suggesting that either aa 161–200 or 201–216 regions is essential for Mis12C recruitment to the CENP-T pathway. Consistent with the Dsn1 levels in these cell lines, both cKO-CT-CC$^{\Delta 73}$/CENP-T$^{\Delta 161-200}$ and cKO-CT-CC$^{\Delta 73}$/CENP-T$^{\Delta 201-216}$ cells died by ~72 h after Tet addition (Fig. 1h). These results suggest that Mis12C binds to the region of aa 161-216 of chicken CENP-T.

cKO-CT-CC$^{\Delta 73}$/CENP-T$^{\Delta 121-240}$ cells died in the presence of Tet (Fig. 1e), which might be due to mitotic defects. To test this prediction, we examined the cell cycle distribution (Fig. 1i), rate of chromosome alignment (Fig. 1j, Supplementary Fig. 1m-o), and the number of spindle poles (Fig. 1k) in these cells in the presence of Tet. Strong accumulation of G2/M fractions was observed (Fig. 1i), and chromosomes were not properly aligned (Fig. 1j, Supplementary Fig. 1o) in cKO-CT-CC$^{\Delta 73}$/CENP-T$^{\Delta 121-240}$ cells. Abnormal numbers of spindle poles were observed (Fig. 1k). These phenotypes were similar to those observed in cKO-CT-CC$^{\Delta 73}$ cells.

We concluded that cKO-CT-CC$^{\Delta 73}$/CENP-T$^{\Delta 121-240}$ cells died due to mitotic defects.

**Ndc80C binding to CENP-T facilitates the Mis12C recruitment to CENP-T.** To examine the functional roles of the KMN network on the CENP-T pathway, it is also important to know how each component of the network is recruited to CENP-T. Previous studies have suggested that CDK1-mediated phosphorylation of CENP-T facilitates recruitment of Ndc80C or Mis12C onto the kinetochore[37,39,41]. We further evaluated the effect of CENP-T phosphorylation on the recruitment of Ndc80C or Mis12C onto CENP-T in cKO-CT-CC$^{\Delta 73}$ cells. We mutated potential CDK1 sites in CENP-T (CENP-T$^{T72A-S88A}$) responsible for Ndc80C binding[39] and introduced a mutant into cKO-CT-CC$^{\Delta 73}$ cells (Fig. 2a, Supplementary Fig. 1f). To evaluate the levels of Ndc80C at kinetochores in these cell lines, we introduced Nuf2-GFP (a component of Ndc80C) into endogenous *Nuf2* alleles (Supplementary Fig. 2a-c). We measured Nuf2-GFP levels at kinetochores. The data indicated that Nuf2 levels were significantly reduced in cKO-CT-CC$^{\Delta 73}$/CENP-T$^{T72A-S88A}$ cells, compared to that in cKO-CT-CC$^{\Delta 73}$/CENP-T$^{WT}$ cells (Fig. 2b). These results support those of our previous study[39].

We also examined the effects of CENP-T phosphorylation on Mis12C recruitment. We further observed eight serine/threonine residues in the aa 161-216 region of chicken CENP-T, and a mutant CENP-T cDNA, in which all serine/threonine residues were replaced with alanine (CENP-T$^{8A}$), was introduced into cKO-CT-CC$^{\Delta 73}$ cells (Fig. 2c, Supplementary Fig. 2d). GFP-Dsn1 was also introduced into the endogenous *Dsn1* alleles in these cells to measure Mis12C levels at the kinetochores (Supplementary Fig. 2d). As shown in Fig. 2d, Mis12C levels in cKO-CT-CC$^{\Delta 73}$/CENP-T$^{8A}$ cells were not significantly different from those in cKO-CT-CC$^{\Delta 73}$/CENP-T$^{WT}$ cells. Consistent with the data of Dsn1 levels, the growth of cKO-CT-CC$^{\Delta 73}$/CENP-T$^{8A}$ cells in the presence of Tet was comparable to that of these cells in the absence of Tet (Fig. 2e). Considering these results, we conclude that phosphorylation of the aa 161–216 region of CENP-T is not essential for the recruitment of Mis12C onto the CENP-T pathway.

Previous CENP-T tethering experiments into a non-centromere locus suggested that the extreme N-terminal region of CENP-T required for Ndc80C binding might be involved in the CENP-T-Mis12C interaction[34,37]. However, it is unclear whether the same regulation occurs in native kinetochores. To examine whether the CENP-T N-terminal region facilitates Mis12C recruitment to CENP-T in native kinetochores, we

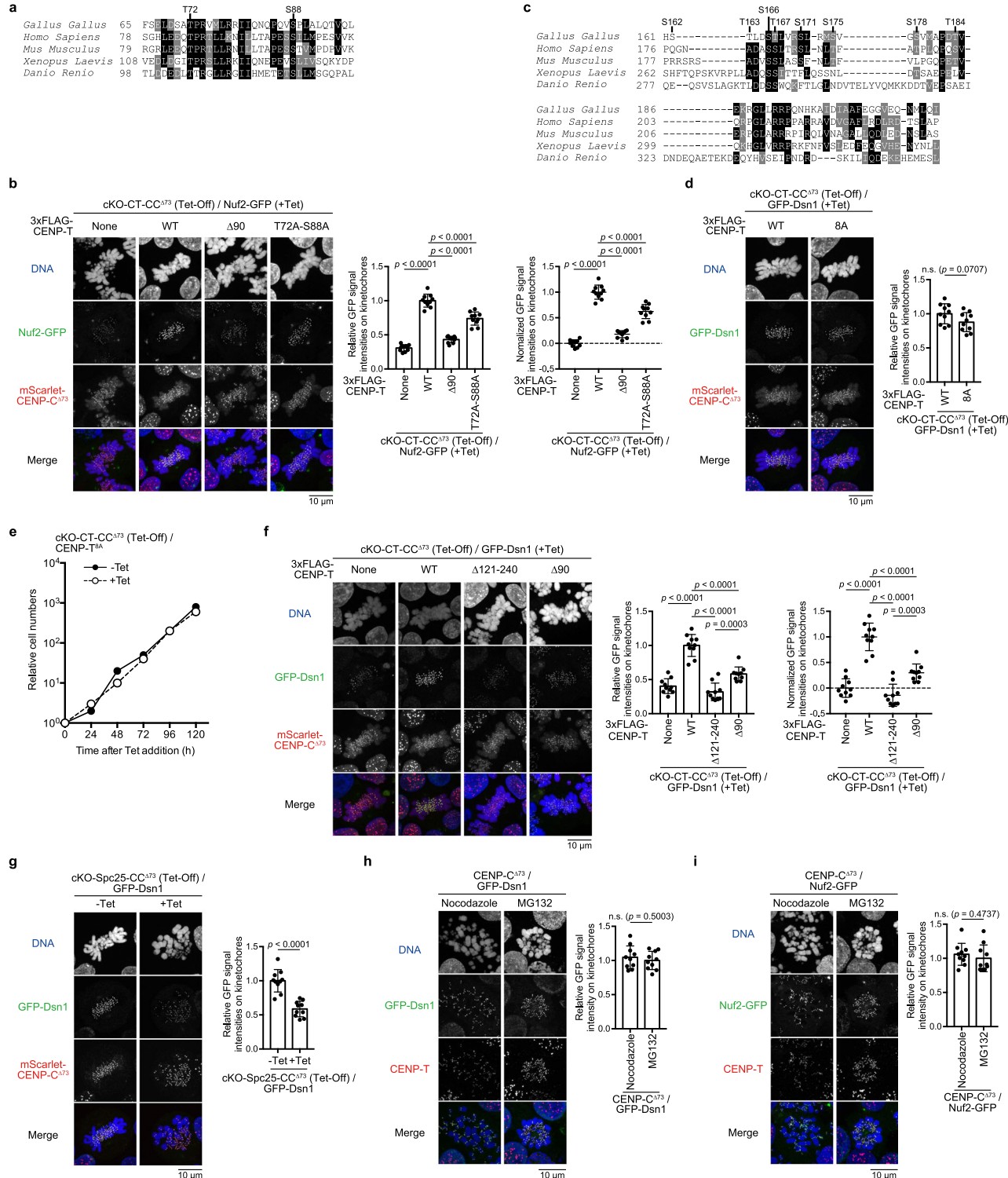

examined Mis12C (Dsn1) levels in cKO-CT-CC$^{\Delta73}$/CENP-T$^{\Delta90}$ cells (Fig. 2f, Supplementary Fig. 2e) and observed that Dsn1 levels in these cells were significantly lower than those in cKO-CT-CC$^{\Delta73}$/CENP-T$^{WT}$ cells (Fig. 2f).

Since the CENP-T N-terminal region is critical for Ndc80C binding, we attempted to demonstrate whether direct binding of Ndc80C to the CENP-T N-terminal region facilitates Mis12C recruitment. We generated cKO-Spc25 (a direct CENP-T-binding protein in Ndc80C) (Tet-Off) cells in the absence of the CENP-C-KMN interaction (cKO-Spc25-CC$^{\Delta73}$ cells) (Supplementary

Fig. 2f-h) in which GFP-Dsn1 was introduced into endogenous *Dsn1* alleles. As shown in Fig. 2g, Dsn1 levels in kinetochores were significantly reduced in cKO-Spc25-CC$^{\Delta73}$ cells after Tet addition, compared with those in these cells before Tet addition. Since these cells contained intact CENP-T, including the Mis12C-binding region, data indicate that direct binding of Ndc80C to CENP-T facilitates Mis12C recruitment onto CENP-T. As Mis12C on CENP-T binds to Knl1C and additional Ndc80C to form a Knl1C-Mis12C-Ndc80C unit (KMN unit), we concluded that direct binding of Ndc80C to CENP-T facilitates the

**Fig. 2 Direct binding of Ndc80C to CENP-T facilitates the recruitment of Mis12C into the kinetochore. a** Amino acids alignment of CENP-T between various species. Potential phosphorylation sites (T72 and S88) are marked. **b** Nuf2-GFP levels at kinetochores in cKO-CENP-T cells expressing mScarlet-CENP-C$^{\Delta73}$ and either CENP-T$^{WT}$, CENP-T$^{\Delta90}$, or CENP-T$^{T72A-S88A}$ in the presence of Tet for 30 h. None indicates parental cKO-CENP-T/CENP-C$^{\Delta73}$/Nuf2-GFP cells. Subtract values are presented using signal intensities of None. DNA was stained with DAPI. Error bars indicate the mean and standard deviation. *p* values were calculated by one-way ANOVA followed by Tukey's test. **c** Amino acids alignment of CENP-T between various species. Mutated S/T residues are marked. **d** GFP-Dsn1 levels at kinetochores in cKO-CENP-T cells expressing mScarlet-CENP-C$^{\Delta73}$ and CENP-T$^{8A}$ in the presence of Tet for 48 h. DNA was stained with DAPI. Error bars indicate the mean and standard deviation. *p* values were calculated by two-tailed Welch's *t*-test. **e** The growth of cKO-CENP-T cells expressing mScarlet-CENP-C$^{\Delta73}$ and either CENP-T$^{8A}$ or CENP-T$^{WT}$ in the absence or presence of Tet. **f** GFP-Dsn1 levels at kinetochores in cKO-CENP-T cells expressing mScarlet-CENP-C$^{\Delta73}$ and either CENP-T$^{WT}$, CENP-T$^{\Delta121-240}$, or CENP-T$^{\Delta90}$ in the presence of Tet for 30 h. None indicates parental cKO-CENP-T/CENP-C$^{\Delta73}$/GFP-Dsn1 cells. Subtract values with None are presented. DNA was stained with DAPI. Error bars indicate the mean and standard deviation. *p* values were calculated as in **b**. **g** GFP-Dsn1 levels at kinetochores in cKO-Spc25 cells expressing mScarlet-CENP-C$^{\Delta73}$ in the absence or presence of Tet for 18 h. DNA was stained with DAPI. Error bars indicate the mean and standard deviation. *p* values were calculated as in **d**. **h** GFP-Dsn1 levels at kinetochores in CENP-C KO cells expressing HA-CENP-C$^{\Delta73}$ (CENP-C$^{\Delta73}$ cells) in the presence of nocodazole or MG132. CENP-T was immuno-stained. DNA was stained with DAPI. Error bars indicate the mean and standard deviation. *p* values were calculated as in **d**. **i** Nuf2-GFP levels at kinetochores in CENP-C$^{\Delta73}$ cells in the presence of nocodazole or MG132. CENP-T was immuno-stained. DNA was stained with DAPI. Error bars indicate the mean and standard deviation. *p* values were calculated as in **d**.

formation of KMN-N (one copy of Ndc80C binds to KM as a KMN unit, and another copy of Ndc80C directly binds CENP-T) on the CENP-T pathway in chicken native kinetochores.

Since Ndc80C-CENP-T direct binding facilitates the Mis12C-CENP-T interaction to form KMN-N, it is possible that the tension caused by kinetochore-microtubule interactions might facilitate Mis12C recruitment onto CENP-T. Using nocodazole (for the absence of tension) or MG132 (for the presence of tension), we prepared chromosome spreads after confirming that chromosomes were aligned in the presence of MG132 (Fig. 2h, i, Supplementary Fig. 2i-m) in CENP-C-deficient cells expressing CENP-C$^{\Delta73}$ (CENP-C$^{\Delta73}$ cells). Further, we compared the levels of Ndc80C and Mis12C at kinetochores in the presence or absence of tension. Both Ndc80C and Mis12C levels did not change in CENP-C$^{\Delta73}$ cells in either the presence or absence of tension (Fig. 2h, i, Supplementary Fig. 2i), suggesting that the tension from microtubules might not have facilitated Mis12C recruitment onto CENP-T. It would be an important further challenge to clarify the molecular mechanisms of how direct binding of Ndc80C to CENP-T facilitates Mis12C recruitment onto CENP-T in cells.

**Knl1C on the CENP-T pathway is essential for sister chromatid cohesion.** Since cKO-CT-CC$^{\Delta73}$/CENP-T$^{\Delta121-240}$ cells led to severe mitotic defects (Fig. 1), we examined the levels of Mis12C-interacting proteins in these cells. We measured Knl1C levels in cells lacking CENP-T-Mis12C and CENP-C-Mis12C interactions (Fig. 3a, Supplementary Fig. 3a). As shown in Fig. 3a, Knl1C levels were significantly reduced in cKO-CT-CC$^{\Delta73}$/CENP-T$^{\Delta121-240}$ cells, and the reduction levels in these cells were similar to those in cKO-CT-CC$^{\Delta73}$ cells.

Since Knl1C recruits Bub1 to kinetochores[10], we investigated Bub1 localization in cKO-CT-CC$^{\Delta73}$/CENP-T$^{\Delta121-240}$ cells, using an anti-Bub1 antibody (Fig. 3b, Supplementary Fig. 3a). Bub1-signals were almost invisible in cKO-CT-CC$^{\Delta73}$/CENP-T$^{\Delta121-240}$ cells, and these levels were similar to those in cKO-CT-CC$^{\Delta73}$ cells (Fig. 3b). Knl1C-Bub1 is a critical target of the checkpoint protein Mps1[11,12], and the checkpoint protein BubR1 also binds to Knl1C[10]. Therefore, localization data of Knl1C and Bub1 suggest that checkpoint functions were compromised in cKO-CT-CC$^{\Delta73}$/CENP-T$^{\Delta121-240}$ cells.

Additionally, Bub1 recruitment leads to the proper localization of the chromosome passenger complex (CPC) and Shugoshin. These complexes together with phosphorylated threonine 3 of histone H3 (H3T3ph) contribute to the maintenance of sister chromatid cohesion in centromeres[10,34,44]. We then examined the distribution of H3T3ph, which is governed by Haspin kinase, in

cKO-CT-CC$^{\Delta73}$/CENP-T$^{\Delta121-240}$ or cKO-CT-CC$^{\Delta73}$/CENP-T$^{WT}$ cells. In cKO-CT-CC$^{\Delta73}$/CENP-T$^{WT}$ cells, H3T3ph signals were concentrated in the centromeric region, but these signals were diffused along the entire chromosome arm in cKO-CT-CC$^{\Delta73}$/CENP-T$^{\Delta121-240}$ cells (Fig. 3c). Consistent with this localization profile of H3T3ph in cKO-CT-CC$^{\Delta73}$/CENP-T$^{\Delta121-240}$ cells, Aurora B (a component of CPC) signals were also distributed in the non-centromeric region in these cells (Fig. 3d). We interpret that the reduction of Knl1C and Bub1 leads to a reduction in the phosphorylation levels of threonine 120 of histone H2A in centromeres governed by Bub1[44,45], which causes the mislocalization of H3T3ph and Aurora B[46,47]. Centromere localization of H3T3ph and Aurora B is related to each other, and the reduction of upstream components affects their centromeric localization (Fig. 3c, d). In addition to these localization profiles, we observed that the distance between sister kinetochores increased in cKO-CT-CC$^{\Delta73}$/CENP-T$^{\Delta121-240}$ cells, compared to that in cKO-CT-CC$^{\Delta73}$/CENP-T$^{WT}$ cells (Fig. 3e). These data suggest that Mis12C and Knl1C (KM) on the CENP-T pathway facilitate proper localization of CPC and H3T3ph to centromeric regions through Bub1 recruitment to maintain the cohesion of sister chromatids (Fig. 3f). KM is known to function on CENP-C[20,48], and we emphasize that our data demonstrated KM functions on CENP-T in cells lacking CENP-C-KMN interaction.

**Two copies of Ndc80C on CENP-T are essential for their proper functions.** While KM (Knl1C and Mis12C) on the CENP-T pathway plays a critical role in the maintenance of sister chromatids cohesion and the recruitment of checkpoint proteins, the roles of Ndc80C must be clarified in the KMN network on CENP-T. CENP-T has two copies of Ndc80C (N-N): one copy directly binds to the CENP-T N-terminus and the second copy is recruited via Mis12C (Fig. 3f). Therefore, the network forms KMN-N. Therefore, it is important to address whether N-N is essential for its microtubule-binding function on the CENP-T pathway independent of KM functions or whether Ndc80C must bind to Mis12C for its function. To test whether N-N is necessary for chromosome segregation, we first examined Ndc80C levels in cells lacking CENP-T-Mis12C and CENP-C-Mis12C interactions. To evaluate Ndc80C levels, we introduced Nuf2-GFP into the endogenous *Nuf2* alleles in cKO-CT-CC$^{\Delta73}$/CENP-T$^{\Delta121-240}$, cKO-CT-CC$^{\Delta73}$/CENP-T$^{\Delta161-200}$, or cKO-CT-CC$^{\Delta73}$/CENP-T$^{\Delta201-216}$ cells (Supplementary Fig. 2a, 3b). As shown in Fig. 4a, Ndc80C levels in these cells were significantly reduced, albeit these levels were higher than those in cKO-CT-CC$^{\Delta73}$ cells (None in Fig. 4a), suggesting that one copy of Ndc80C still binds to the

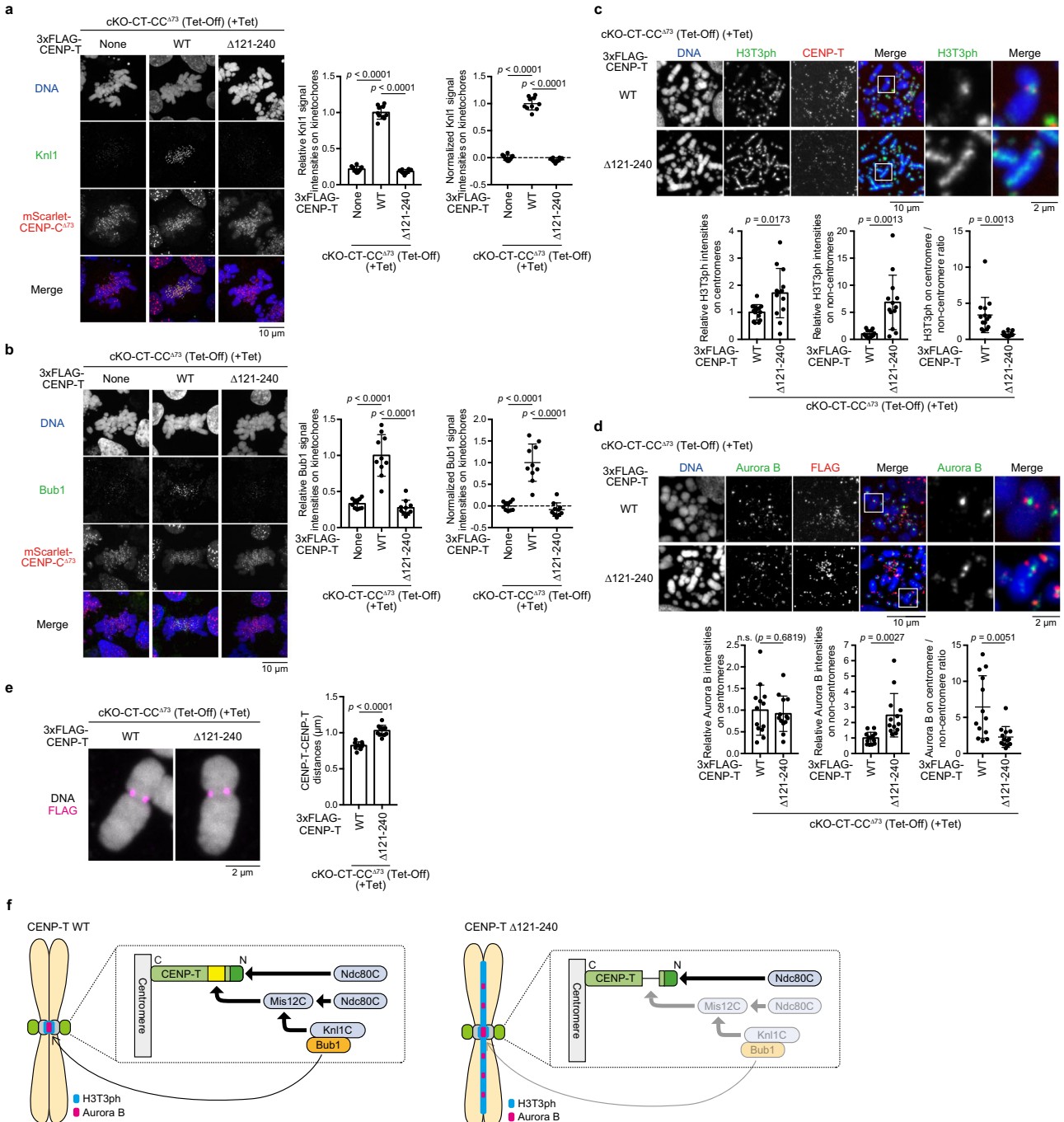

**Fig. 3 Reduction of Knl1C levels and sister chromatid cohesion defects in cells lacking the CENP-T-Mis12C interaction. a** Knl1 levels at kinetochores in cKO-CENP-T cells expressing mScarlet-CENP-C$^{\Delta73}$ and either CENP-T$^{WT}$ or CENP-T$^{\Delta121-240}$ in the presence of Tet for 30 h. None indicates parental cKO-CENP-T/CENP-C$^{\Delta73}$ cells. Subtract values are presented using signal intensities of None. DNA was stained with DAPI. Error bars indicate the mean and standard deviation. *p* values were calculated by one-way ANOVA followed by Tukey's test. **b** Bub1 levels at kinetochores in cKO-CENP-T cells expressing mScarlet-CENP-C$^{\Delta73}$ and either CENP-T$^{WT}$ or CENP-T$^{\Delta121-240}$ in the presence of Tet for 30 h. None indicates parental cKO-CENP-T/CENP-C$^{\Delta73}$ cells. Subtract values with None are presented. DNA was stained with DAPI. Error bars indicate the mean and standard deviation. *p* values were calculated as in **a**. **c** Immuno-localization of H3T3ph on chromosomes in cKO-CENP-T cells expressing mScarlet-CENP-C$^{\Delta73}$ and either CENP-T$^{WT}$ or CENP-T$^{\Delta121-240}$ in the presence of Tet for 30 h and nocodazole for last 4 h. The insets show magnified views of the boxed regions. Graphs summarize H3T3ph localization profiles on centromeres or non-centromeres. Error bars show the mean and standard deviation. *p* values were calculated by two-tailed Welch's *t*-test. **d** Immuno-localization of Aurora B on chromosomes in cKO-CENP-T cells expressing mScarlet-CENP-C$^{\Delta73}$ and either CENP-T$^{WT}$ or CENP-T$^{\Delta121-240}$ in the presence of Tet for 30 h and nocodazole for last 4 h. The insets present magnified views. Graphs summarize Aurora B localization profiles on centromeres or non-centromeres. Error bars show the mean and standard deviation. *p* values were calculated as in **c**. **e** Kinetochore-kinetochore distances of mitotic chromosomes in cKO-CENP-T cells expressing mScarlet-CENP-C$^{\Delta73}$ and either CENP-T$^{WT}$ or CENP-T$^{\Delta121-240}$ in the presence of Tet for 30 h and nocodazole for last 4 h. Error bars indicate the mean and standard deviation. *p* values were calculated as in **c**. **f** Mis12C and Knl1C (KM) on the CENP-T pathway function for the maintenance of sister chromatid cohesion (left). When KM is absent from CENP-T (right), H3T3ph and Aurora B are diffused into non-centromere regions, causing defects in sister chromatid cohesion.

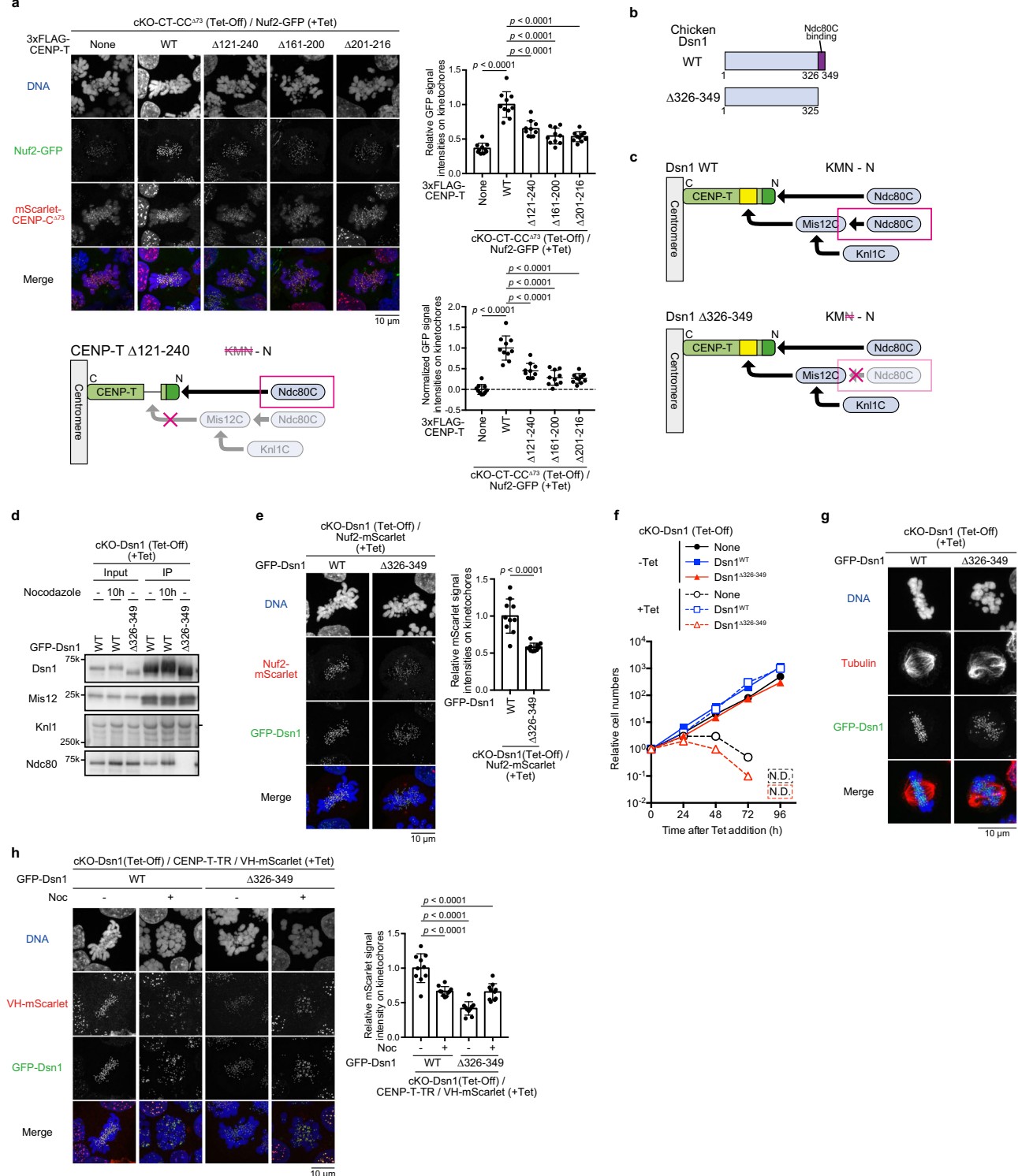

N-terminal region of CENP-T$^{\Delta 121-240}$ (Fig. 4a), which is consistent with our previous observation[35].

Although we observed that one copy of Ndc80C per one CENP-T still exists in cells lacking Mis12C, we analyzed the impact of one copy of Ndc80C without disrupting other Mis12C functions. Previous studies have suggested that the Dsn1 C-terminal region in Mis12C is critical for Ndc80C binding (Fig. 4b, c)[35,48]. Although another study suggested that the PVIHL motif in human Nsl1 in Mis12C might be also involved in Ndc80C binding[43], this region in Nsl1 was not essential in DT40

cells, based on analyses using cKO-Nsl1 (Auxin-Inducible-Degron: AID) cells (Supplementary Fig. 4a-e). Therefore, in this study, we focused on the Dsn1 C-terminal region. We generated cKO-Dsn1 (Tet-Off) cells, in which GFP-Dsn1$^{WT}$ (cKO-Dsn1/Dsn1$^{WT}$) or GFP-Dsn1$^{\Delta 326-349}$ lacking the Ndc80C-binding region (cKO-Dsn1/Dsn1$^{\Delta 326-349}$) were expressed under the control of the endogenous Dsn1 promoter (Supplementary Fig. 5a). We confirmed that Dsn1$^{\Delta 326-349}$ formed Mis12C and bound to Knl1C but not to Ndc80C based on immunoprecipitation experiments using an anti-GFP nanobody (Fig. 4d).

**Fig. 4 Two copies of Ndc80C (N-N) are essential for proper kinetochore-microtubules interaction. a** Nuf2-GFP levels at kinetochores in cKO-CENP-T cells expressing mScarlet-CENP-C$^{\Delta 73}$ and various CENP-T mutants in the presence of Tet for 30 h. None indicates parental cKO-CENP-T/CENP-C$^{\Delta 73}$/ Nuf2-GFP cells. Subtract values were presented using signal intensities of None. DNA was stained with DAPI. Error bars show the mean and standard deviation. *p* values were calculated by one-way ANOVA followed by Tukey's test. **b** Schematic representation of chicken Dsn1$^{\Delta 326-349}$. **c** Schematic representation explaining that chicken Dsn1$^{\Delta 326-349}$ does not bind to Ndc80C but binds to Knl1C via the formation of Mis12C. **d** Immunoprecipitation with anti-GFP in cKO-Dsn1 cells expressing either GFP-Dsn1$^{WT}$ or GFP-Dsn1$^{\Delta 326-349}$ in the presence of Tet. Samples were subjected to immunoblot analyses using anti-Dsn1, -Mis12, -Knl1, and -Ndc80 antibodies. **e** Nuf2-mScarlet levels at kinetochores in cKO-Dsn1 cells expressing either GFP-Dsn1$^{WT}$ or GFP-Dsn1$^{\Delta 326-349}$ in the presence of Tet for 30 h. DNA was stained with DAPI. Error bars indicate the mean and standard deviation. *p* values were calculated by two-tailed Welch's t-test. **f** The growth of cKO-Dsn1 cells expressing either GFP-Dsn1$^{WT}$ or GFP-Dsn1$^{\Delta 326-349}$ in the absence or presence of Tet (–Tet or +Tet). None indicates parental cKO-Dsn1 cells. **g** Microtubule staining using an anti-tubulin antibody (red) for cKO-Dsn1 cells expressing either GFP-Dsn1$^{WT}$ or GFP-Dsn1$^{\Delta 326-349}$ in the presence of Tet for 30 h. The cells were incubated in ice-cold medium for 10 min prior to fixation. DNA was stained with DAPI. **h** mScarlet-fused vinculin head domain (VH-mScarlet) localization to kinetochore in cKO-Dsn1 cells expressing CENP-T-TR reporter and either GFP-Dsn1 $^{WT}$ or GFP-Dsn1$^{\Delta 326-349}$. Disordered region of CENP-T (aa 241-529) was exchanged with the chicken TR domain (aa 482-889) (CENP-T TR), was expressed from the CENP-T locus. Cells were treated with or without nocodazole (Noc + or –), and mScarlet signals were measured. DNA was stained with DAPI. Error bars indicate the mean and standard deviation. *p* values were calculated as in **a**.

Consistent with the biochemical analyses, Dsn1 and Knl1 levels at kinetochores in cKO-Dsn1/Dsn1$^{\Delta 326-349}$ cells were more than 80%, compared with those in cKO-Dsn1/Dsn1$^{WT}$ cells (Supplementary Fig. 5a-c), and Bub1 levels were similar to those in control cells (Supplementary Fig. 5a, d). In addition, the localization profiles of H3T3ph and Aurora B were similar to those in control cells (Supplementary Fig. 5e, f); therefore, sister chromatid cohesion defects were not observed (Supplementary Fig. 5g). These results suggest that KM retains the checkpoint function and sister chromatid cohesion maintenance function, lacking KM and Ndc80C interaction in cKO-Dsn1/Dsn1$^{\Delta 326-349}$ cells (KM-N).

In contrast, Ndc80C (Nuf2-mScarlet) levels in cKO-Dsn1/ Dsn1$^{\Delta 326-349}$ cells were significantly reduced, albeit Nuf2 signals were still visible (Fig. 4e, Supplementary Fig. 5h), suggesting that this mutation does not affect the CENP-T-Ndc80C direct interaction. We further investigated the growth rate of cKO-Dsn1/Dsn1$^{\Delta 326-349}$ cells and observed that these cells died by ~ 96 h after Tet addition (Fig. 4f). Although cKO-Dsn1/Dsn1$^{\Delta 326-349}$ cells died, kinetochores in these cells may still be bound to microtubules, since some of the Ndc80C are still bound to CENP-T. We then stained microtubules and observed chromosome alignments in cKO-Dsn1/Dsn1$^{\Delta 326-349}$ cells. As shown in Fig. 4g, unaligned chromosomes and abnormal spindles were observed in these cells, and microtubules did not appear to bind to kinetochores. These data indicate that proper kinetochore-microtubule attachment does not occur in cKO-Dsn1/Dsn1$^{\Delta 326-349}$ cells. Consistent with microtubule staining, tension from microtubules was not applied to CENP-T in these cells, based on analyses using the tension sensor probe (Fig. 4h, Supplementary Fig. 4i, j). We used the talin rod (TR) tension sensor system[49,50], similar to that used in our previous study[35]. We expressed vinculin head domain (VH)-mScarlet and CENP-T-TR, and VH-mScarlet localized to kinetochores via binding to CENP-T-TR. In cells with wild-type Dsn1, VH-mScarlet signals decreased upon nocodazole treatment. However, in cells expressing Dsn1$^{\Delta 326-349}$ VH-mScarlet signals did not decrease upon nocodazole treatment (Fig. 4h). Considering these results, one copy of Ndc80C directly binding to CENP-T is not sufficient for its microtubule-binding function, suggesting that two copies of Ndc80C (N-N) on CENP-T are essential for the establishment of appropriate kinetochore-microtubule interaction.

**KM-Ndc80C interaction is not essential, and they are separable.** Since some KM roles were ensured without interaction with Ndc80C (Fig. 5a), KM and Ndc80C might not be necessary to interact with each other and are separable (Fig. 5a). Alternatively, it is still possible that a second copy of Ndc80C might be necessary to interact with KM for its functions. To distinguish

between these possibilities, we engineered the CENP-T N-terminus. If we introduced an additional Ndc80C-binding domain (aa 1-90) in the CENP-T N-terminal region, extra Ndc80C should be recruited without Mis12C binding (Fig. 5a, b). Initially, we introduced CENP-T$^{2x(1-90)}$ into the endogenous *CENP-T* allele in cKO-CT-CC$^{\Delta 73}$ cells and confirmed that CENP-T$^{2x(1-90)}$ functions as wild-type CENP-T by examining the growth of these cells in the presence of Tet (Supplementary Fig. 6a, b). We also observed an increase in Ndc80C (Nuf2-GFP) levels in cKO-CT-CC$^{\Delta 73}$/CENP-T$^{2x(1-90)}$ cells owing to an additional Ndc80C-binding site on CENP-T (Fig. 5c).

As CENP-T$^{\Delta 121-240}$ lacks the binding of Mis12C, which binds to Knl1C and Ndc80C, it loses KM and one Ndc80C copy. Therefore, in addition to defects due to a lack of KM roles (Fig. 3), microtubule-binding defects owing to Ndc80C reduction could occur in cKO-CT-CC$^{\Delta 73}$/CENP-T$^{\Delta 121-240}$ cells (Fig. 4a). We further stained microtubules and observed chromosome alignments in cKO-CT-CC$^{\Delta 73}$/CENP-T$^{\Delta 121-240}$, cKO-CT-CC$^{\Delta 73}$/ CENP-T$^{WT}$, or cKO-CT-CC$^{\Delta 73}$ cells (Fig. 5d). We frequently observed unaligned chromosomes, and the spindle shape was not normal in cKO-CT-CC$^{\Delta 73}$/CENP-T$^{\Delta 121-240}$ cells, as observed in cKO-CT-CC$^{\Delta 73}$ cells (Fig. 5d). We also observed abnormal numbers of spindle poles (Fig. 5e), a higher rate of misalignment of chromosomes (Fig. 5f), and an increase in G2/M populations (Fig. 5g) in cKO-CT-CC$^{\Delta 73}$/CENP-T$^{\Delta 121-240}$ cells. These data suggest that proper kinetochore-microtubule attachment and spindle integrity were disrupted in cKO-CT-CC$^{\Delta 73}$/CENP-T$^{\Delta 121-240}$ cells. However, the addition of the Ndc80C-binding site to CENP-T$^{\Delta 121-240}$ (CENP-T$^{2x(1-90)\_\Delta 121-240}$) might rescue defects in proper kinetochore-microtubule attachments. We further generated cKO-CT-CC$^{\Delta 73}$/CENP-T$^{2x(1-90)\_\Delta 121-240}$ cells (Supplementary Fig. 6c) and analyzed the phenotype of these cells. Abnormal spindle shape, improper kinetochore-microtubule attachments, number of spindle poles, and defects in chromosome misalignment were largely suppressed in cKO-CT-CC$^{\Delta 73}$/CENP-T$^{2x(1-90)\_\Delta 121-240}$ cells, although recovery of chromosome misalignment was not strong, compared with cKO-CT-CC$^{\Delta 73}$/CENP-T$^{WT}$ cells (Fig. 5d–f, Supplementary Fig. 6d). These data suggest that some Ndc80C-related defects in cKO-CT-CC$^{\Delta 73}$/CENP-T$^{\Delta 121-240}$ cells were partially suppressed by the addition of extra Ndc80C-binding sites. However, significant cohesion defects were still observed in cKO-CT-CC$^{\Delta 73}$/ CENP-T$^{2x(1-90)\_\Delta 121-240}$ cells (Fig. 5h). In addition, cKO-CT-CC$^{\Delta 73}$/ CENP-T$^{2x(1-90)\_\Delta 121-240}$ cells still revealed G2/M accumulation (Fig. 5g) and ultimately died by ~ 96 h after Tet addition (Fig. 5i), implying that the addition of extra Ndc80C to CENP-T did not completely suppress mitotic defects in cKO-CT-CC$^{\Delta 73}$/CENP-T$^{\Delta 121-240}$ cells. One possible explanation is that KM mitotic functions are essential for cell growth. Another possible explanation is that the

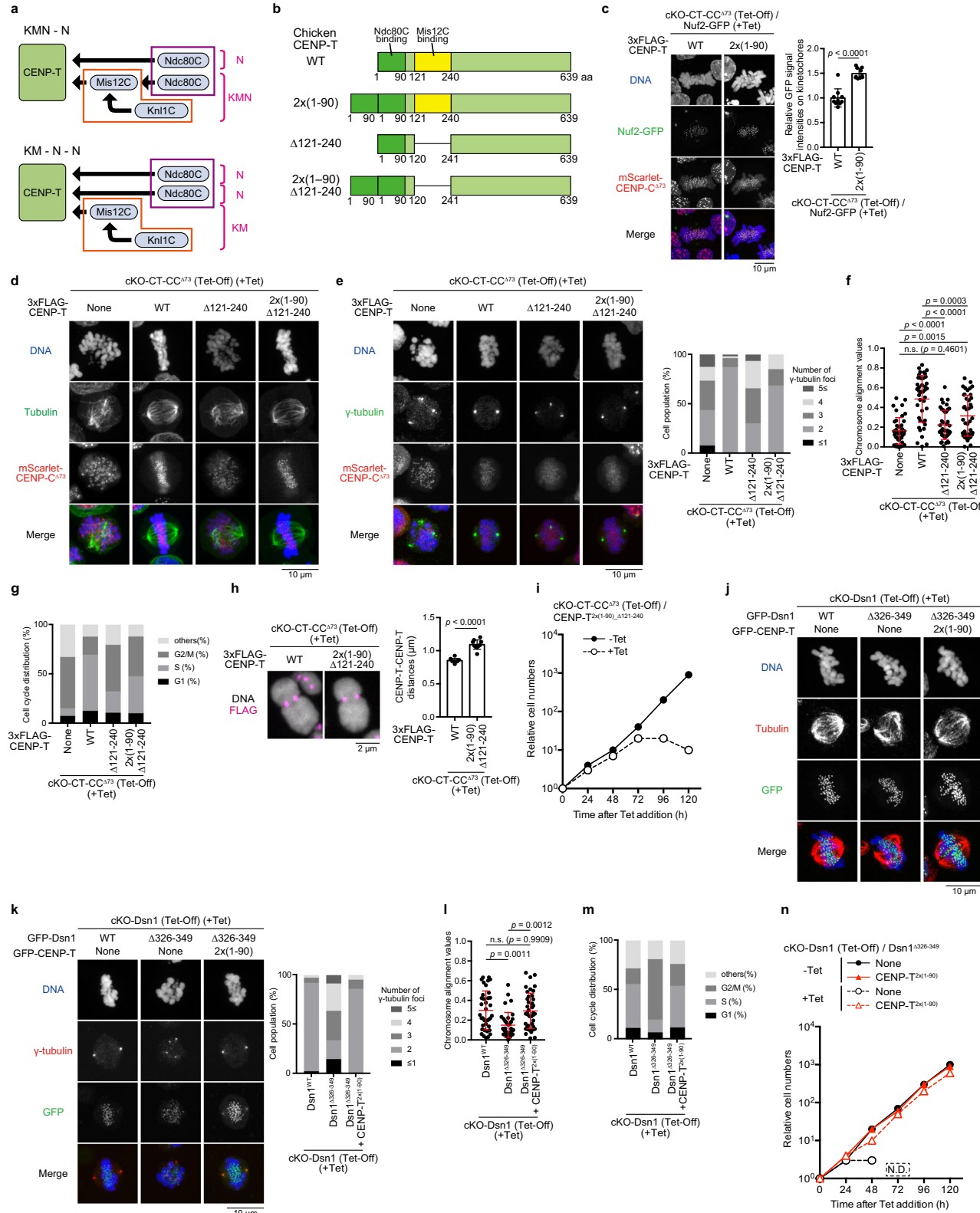

Mis12C-Ndc80C interaction is still critical for full mitotic functions and cell growth, even if the defects of microtubule-kinetochore attachment were rescued by extra Ndc80C.

To distinguish between these two possibilities, we expressed CENP-T$^{2x(1-90)}$ in cKO-Dsn1/Dsn1$^{\Delta326-349}$ cells (Supplementary Fig. 6e), which lost one copy of Ndc80C and retained KM functions (KM-N). Although the KMN network forms KMN-N

(one copy of Ndc80C binds to Mis12C, and another Ndc80C directly binds CENP-T) in wild-type chicken cells, KM-N-N (two copies of Ndc80C directly bind to CENP-T distinct from KM) may be functional (Fig. 5a). Strikingly, expression of GFP-CENP-T$^{2x(1-90)}$ in cKO-Dsn1/Dsn1$^{\Delta326-349}$ cells completely compensated for mitotic defects in the cKO-Dsn1/Dsn1$^{\Delta326-349}$ cells in the presence of Tet (Fig. 5j-m, Supplementary Fig. 6f). Abnormal

**Fig. 5 KM and N-N are separable in native kinetochores. a** The artificial conversion of KMN-N to KM-N-N. **b** Various CENP-T constructs providing an extra Ndc80C-binding site. **c** Nuf2-GFP levels at kinetochores in cKO-CENP-T cells expressing mScarlet-CENP-C$^{\Delta 73}$ and either CENP-T$^{WT}$ or CENP-T$^{2X(1-90)}$ in the presence of Tet for 30 h. DNA was stained with DAPI. Error bars show the mean and standard deviation. *p* values were calculated by two-tailed Welch's *t*-test. **d** Tubulin staining in cKO-CENP-T cells expressing mScarlet-CENP-C$^{\Delta 73}$ and either CENP-T$^{WT}$, CENP-T$^{\Delta 121-240}$, or CENP-T$^{2X(1-90)\_\Delta 121-240}$ in the presence of Tet for 30 h. None indicates CENP-T KO cells. **e** Number distributions of spindle poles stained with anti-γ-tubulin in cKO-CENP-T/CENP-C$^{\Delta 73}$ (None), cKO-CENP-T/CENP-C$^{\Delta 73}$ expressing CENP-T$^{WT}$, CENP-T$^{\Delta 121-240}$, or CENP-T$^{2X(1-90)\_\Delta 121-240}$. **f** Chromosome alignment values in cKO-CENP-T/CENP-C$^{\Delta 73}$ (None), cKO-CENP-T/CENP-C$^{\Delta 73}$ expressing CENP-T$^{WT}$, CENP-T$^{\Delta 121-240}$, or CENP-T$^{2X(1-90)\_\Delta 121-240}$ in the presence of Tet for 30 h and MG132 for last 3 h. Error bars indicate the mean and standard deviation. *p* values were calculated by one-way ANOVA followed by Tukey's test. **g** Cell cycle distribution of cKO-CENP-T/CENP-C$^{\Delta 73}$ (None), cKO-CENP-T/CENP-C$^{\Delta 73}$ expressing CENP-T$^{WT}$, CENP-T$^{\Delta 121-240}$, or CENP-T$^{2X(1-90)\_\Delta 121-240}$. **h** Kinetochore-kinetochore distances in cKO-CENP-T cells expressing mScarlet-CENP-C$^{\Delta 73}$ and either CENP-T$^{WT}$ or CENP-T$^{2X(1-90)\_\Delta 121-240}$ in the presence of Tet for 30 h. Error bars indicate the mean and standard deviation. *p* values were calculated as in **c**. **i** The growth of cKO-CENP-T cells expressing mScarlet-CENP-C$^{\Delta 73}$ and CENP-T$^{2X(1-90)\_\Delta 121-240}$ in the absence or presence of Tet. **j** Tubulin staining in cKO-Dsn1 cells expressing Dsn1$^{WT}$, Dsn1$^{\Delta 326-349}$, or both Dsn1$^{\Delta 326-349}$ and CENP-T$^{2X(1-90)}$ in the presence of Tet for 30 h. **k** Number distributions of spindle poles in cKO-Dsn1 cells expressing Dsn1$^{WT}$, Dsn1$^{\Delta 326-349}$ or both Dsn1$^{\Delta 326-349}$ and CENP-T$^{2X(1-90)}$. **l** Chromosome alignment values in cKO-Dsn1 cells expressing Dsn1$^{WT}$, Dsn1$^{\Delta 326-349}$ or both Dsn1$^{\Delta 326-349}$ and CENP-T$^{2X(1-90)}$ in the presence of Tet for 30 h and MG132 for last 3 h. Error bars indicate the mean and standard deviation. *p* values were calculated as in **f**. **m** Cell cycle distribution of cKO-Dsn1 cells expressing Dsn1$^{WT}$, Dsn1$^{\Delta 326-349}$ or both Dsn1$^{\Delta 326-349}$ and CENP-T$^{2X(1-90)}$. **n** The growth of cKO-Dsn1 cells expressing both GFP-Dsn1$^{\Delta 326-349}$ and 3XFLAG-CENP-T$^{2X(1-90)}$ in the absence or presence of Tet. None indicates no extra CENP-T expression.

spindle shape, abnormal numbers of spindle poles, mitotic accumulation, and chromosome misalignment were almost suppressed. Additionally, growth defects in cKO-Dsn1/Dsn1$^{\Delta 326-349}$ cells were completely rescued by the expression of GFP-CENP-T$^{2x(1-90)}$ (Fig. 5n). These data indicate that two copies of Ndc80C can be functional without direct binding to Mis12C. This result indicates that direct binding of Ndc80C to Mis12C is not required for the microtubule-binding function of Ndc80C. Comparing the phenotype of cKO-Dsn1/Dsn1$^{\Delta 326-349}$ cells expressing CENP-T$^{2x(1-90)}$ (Fig. 5j-n) with that of cKO-CT-CC$^{\Delta 73}$/CENP-T$^{2x(1-90)\_\Delta 121-240}$ cells (Fig. 5d-i), we demonstrated that KM has essential mitotic roles, distinct from the microtubule-binding functions of N-N. We conclude that KM and N-N (two copies of Ndc80C) are separable and have distinct essential functions for cell growth (Fig. 5a).

**Mis12C-Ndc80C interaction is dispensable in human RPE-1 cells**. Our analyses in chicken DT40 cells demonstrated that KMN-N can be converted to KM-N-N in the KMN network on the CENP-T pathway (Fig. 6a). Interestingly, human CENP-T has three Ndc80C-binding sites (two are direct binding sites and one is via Mis12C: KMN-N-N) (Fig. 6a)[35,41,51]. If two copies of Ndc80C on CENP-T are sufficient for microtubule-kinetochore interactions in human cells, hDsn1 knockout cells expressing a mutant hDsn1 without the Ndc80C-binding site would be viable (KM-N-N), unlike cKO-Dsn1/Dsn1$^{\Delta 326-349}$ DT40 cells. To test this hypothesis, we generated cKO-hDsn1 RPE-1 cells based on the AID system (cKO-hDsn1 (AID)) and expressed hDsn1$^{\Delta 325-356}$ lacking the Ndc80C-binding region in these cells (Supplementary Fig. 7a-f). As observed in cKO-Dsn1/Dsn1$^{\Delta 326-349}$ chicken DT40 cells, human Dsn1$^{\Delta 325-356}$ was not associated with Ndc80C based on an immunoprecipitation experiment (Fig. 6b). In addition, Ndc80C levels were significantly decreased in cKO-hDsn1/hDsn1$^{\Delta 325-356}$ cells after Indole-3-acetic acid (IAA) addition (Fig. 6c). Further, we examined the mitotic phenotype (Fig. 6d-f). In cKO-hDsn1 cells after IAA addition, cells with chromosome misalignment and lagging chromosomes significantly increased (Fig. 6e, f). However, the expression of Dsn1$^{WT}$ and Dsn1$^{\Delta 325-356}$ suppressed these mitotic defects (Fig. 6d-f). In addition, the growth rate of cKO-hDsn1 cells expressing Dsn1$^{\Delta 325-356}$ was comparable to those of cKO-hDsn1/hDsn1$^{WT}$ cells (Fig. 6g). These data support our hypothesis. Thus, we conclude that the Mis12C-Ndc80C interaction is dispensable in human cells.

## Discussion

We previously revealed that the CENP-T pathway is the main pathway for the recruitment of the KMN network onto kinetochores, compared to the CENP-C pathway, in chicken DT40 cells[35]. However, the specific roles of the KMN network on the CENP-T pathway remain unclear. The KMN network binds to both CENP-C and CENP-T. Therefore, to precisely evaluate the roles of the KMN network on the CENP-T pathway, we must remove the CENP-C-KMN interaction to eliminate some redundant functions of CENP-C. Moreover, since some proteins have multiple roles, we must use specific mutants of each protein, which do not disrupt their multiple functions to avoid misinterpretation of its protein function. Based on our analyses, we concluded that the KMN network forms KMN-N in chicken cells and that KM and N-N have distinct roles: Proper kinetochore-microtubule interaction via N-N (two copies of Ndc80C) and the maintenance of sister chromatid cohesion, recruitment of one copy of Ndc80C or checkpoint function via KM (Knl1C and Mis12C) (Fig. 7a, b). Importantly, we observed that the Mis12C-Ndc80C interaction was not necessary, and KM and N-N are separable (Fig. 7a, b). Since the Ndc80C-Mis12C interaction is thought to be critical for the kinetochore functions[7,42,43], our observations provide new insights for kinetochore studies.

There are multiple recruitment pathways of the KMN network onto kinetochores in other organisms, in addition to the CENP-C and CENP-T-pathways[52,53], and the pathway choice is variable among different species[54]. Although many species have more than two pathways, *Drosophila melanogaster* and *Caenorhabditis elegans* have only the CENP-C pathway[55-57]. In contrast, several Lepidoptera and Mucor have only the CENP-T pathway[58,59]. These observations suggest that each pathway can recruit the KMN network onto the kinetochore, and the CENP-T and CENP-C pathways might be redundant in organisms possessing more than two pathways. However, our previous study indicated that the CENP-T-KMN network interaction is essential, but the CENP-C-KMN network interaction is dispensable in chicken DT40 cells. Nevertheless, we hypothesized that the CENP-C-KMN network interaction plays a role, because cells lacking Mis12C binding to CENP-T in the presence of the CENP-C-KMN network interaction do not completely die. This study demonstrated that cells completely died in the absence of Mis12C interaction with both CENP-C and CENP-T, suggesting that Mis12C on CENP-C plays a role, which might contribute to sister chromatid cohesion or checkpoint function via KM or robust kinetochore-microtubule attachment via additional Ndc80C since

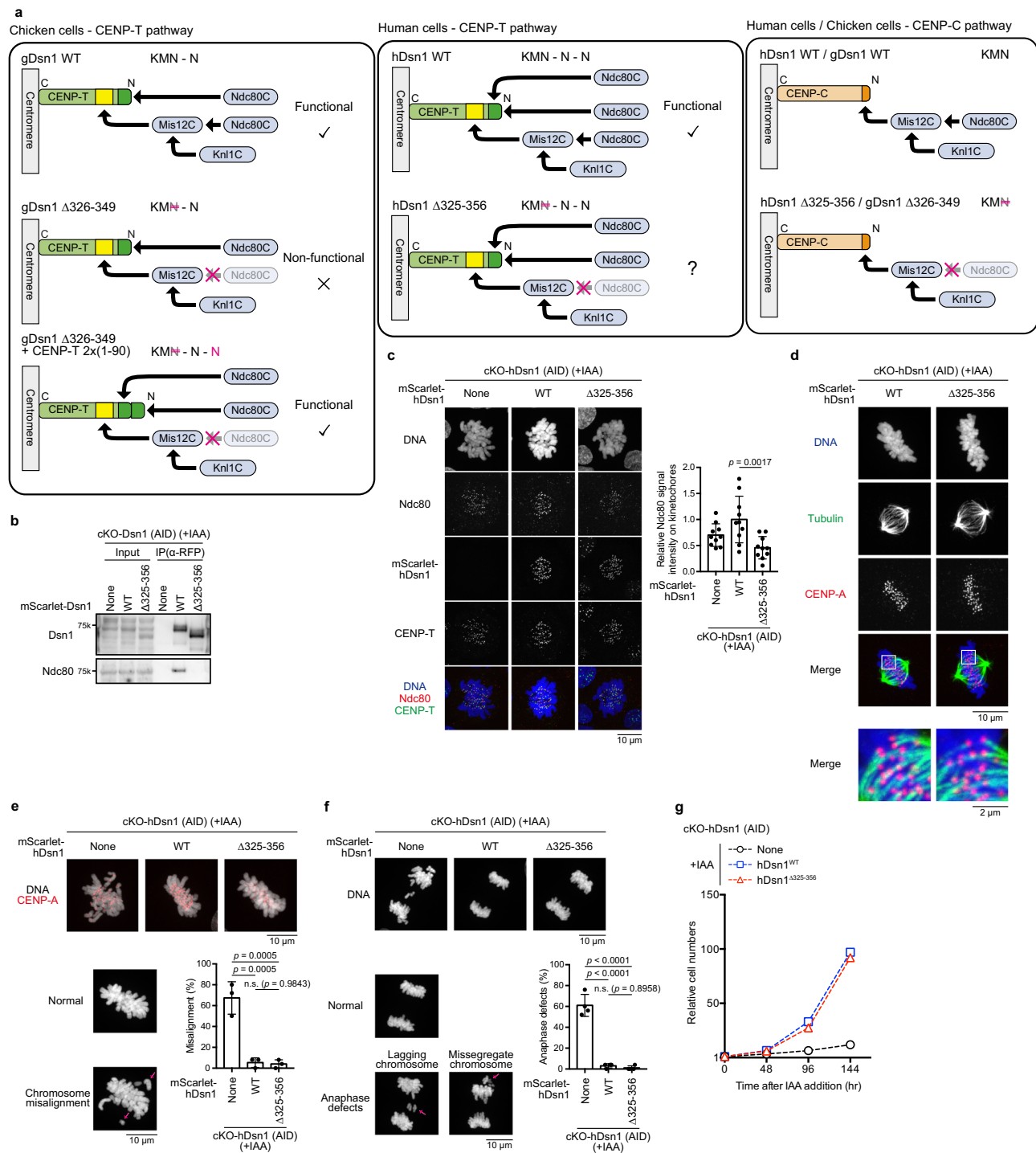

Mis12C binds to Knl1C and Ndc80C. Therefore, to analyze the functional roles of KMN in each pathway, we must characterize it with the depletion of some redundancies. This strategy is essential for evaluating the function of each complex in different organisms with various recruitment pathways of the KMN network.

Two copies of Ndc80C (N-N) are necessary and sufficient for proper kinetochore-microtubule attachment in DT40 cells, suggesting that binding of a single copy of Ndc80C to spindle microtubules is not sufficient for full kinetochore functions. Single-molecule experiments demonstrated that the multivalency of Ndc80C efficiently tracked depolymerizing microtubules, whereas a single copy of Ndc80C did not[51], which supports our observations. Other biophysics experiments suggest that

microtubule pulling forces from kinetochores vary (0.4–8 pN)[60,61], suggesting that the double increase of microtubule-binding proteins might not be sufficient for stable chromosome segregation. However, it is unclear how much the kinetochore with one copy of Ndc80C is unstable, and such kinetochores might not generate sufficient force for chromosome segregation. Thus, the extra copy of Ndc80C might have synergistic effects on kinetochore function in vivo. It is important to know how extra copies of Ndc80C contribute to kinetochore function.

The CENP-C-KMN network interaction is not sufficient in DT40 cells since Mis12C is easily dissociated from CENP-C[35]. However, forced binding of Mis12C to CENP-C stabilizes the CENP-C pathway. In this situation, there is only one copy of

**Fig. 6 Mis12C-Ndc80C interaction is dispensable in human cells. a** Schematic representation of the KMN network on CENP-T in chicken and human cells. Chicken has KMN-N on CENP-T. While cKO-Dsn1/Dsn1$^{\Delta326-349}$ cells (KMN-N) died, expression of CENP-T$^{2X(1-90)}$ in these cells (KMN-N-N) rescued growth deficiency. In human cells, KMN-N-N is on CENP-T. The phenotype of cKO-Dsn1 lacking the Mis12C-Ndc80C interaction (cKO-hDsn1/hDsn1$^{\Delta325-356}$ cell: KMN-N-N) will be tested. The KMN network on the CENP-C pathway in chicken and human cells is also presented. Both gDsn1$^{\Delta326-349}$ and hDsn1$^{\Delta325-356}$ do not bind to Ndc80C on the CENP-C pathway. **b** Immunoprecipitation with anti-RFP in AID-based cKO-hDsn1 cells expressing either mScarlet-hDsn1$^{WT}$ or -hDsn1$^{\Delta325-356}$ in the presence of nocodazole for 24 h and IAA for last 2 h. Samples were analyzed by anti-Dsn1, and -Ndc80 antibodies. **c** Ndc80 levels at kinetochores in cKO-hDsn1 cells expressing mScarlet-hDsn1$^{WT}$ or -hDsn1$^{\Delta325-356}$ in the presence of IAA for 2 h. DNA was stained with DAPI. Ndc80 was stained with an anti-Ndc80 antibody (red). CENP-T was stained with an anti-CENP-T antibody (green). None indicates parental cKO-hDsn1 (AID) cells. Error bars indicate the mean and standard deviation. $p$ values were calculated by one-way ANOVA followed by Tukey's test. **d** Tubulin staining in cKO-hDsn1 cells expressing hDsn1$^{WT}$ or hDsn1$^{\Delta325-356}$ in the presence of IAA for 2 h. Kinetochores were stained with an anti-human CENP-A antibody (red). The insets present the kinetochore-microtubule bindings. DNA was stained with DAPI. **e** Images of mitotic chromosomes stained with anti-CENP-A in cKO-hDsn1 cells (None) expressing hDsn1$^{WT}$ or hDsn1$^{\Delta325-356}$ in the presence of IAA for 2 h. DNA was stained with DAPI. The graph summarizes the percentage of cells with misaligned chromosomes in each line. Error bars indicate the mean and standard deviation. $p$ values were calculated as in **c**. **f** Images of anaphase for cKO-hDsn1 cells (None) expressing hDsn1$^{WT}$ or hDsn1$^{\Delta325-356}$ in the presence of IAA for 2 h. DNA was stained with DAPI. The graph summarizes the percentage of cells with abnormal anaphase in each line. Error bars indicate the mean and standard deviation. $p$ values were calculated as in **c**. **g** The growth of cKO-hDsn1 cells expressing mScarlet-hDsn1$^{WT}$ or -hDsn1$^{\Delta325-356}$ in the absence or presence of IAA.

Ndc80C on the kinetochore, which possesses only the CENP-C pathway. However, kinetochores with one copy of the Ndc80C were functional in this case. This mechanism might be explained by the amount of protein at the kinetochore[38]. Suzuki et al.[38] proposed ~215 molecules of CENP-C and ~72 molecules of CENP-T in a human kinetochore. Since only ~70 molecules of Mis12C bind to CENP-C, two-thirds of CENP-C are usually not used for the KMN network interaction. However, CENP-C has the capacity to accept more KMN components, and forced binding of Mis12C to CENP-C would cause a stable CENP-C pathway, even if the kinetochore with the CENP-C pathway has only one copy of Ndc80C. Additionally, this measurement may explain why two copies of Ndc80C are necessary for CENP-T. Compared with CENP-C amounts, CENP-T amounts are small at each kinetochore[38]. To maintain sufficient amounts of Ndc80C, two copies of Ndc80C per CENP-T may be critical for chromosome segregation.

Independent of the microtubule-binding function of two copies of Ndc80C (N-N), Knl1C and Mis12C (KM) have some essential roles, since even when cells had two copies of Ndc80C for proper kinetochore-microtubule interaction in cKO-CT-CC$^{\Delta73}$/CENP-T$^{2x(1-90)\_\Delta121-240}$ cells, the cells died. Although it is still unclear which KM functions are critical for cell viability, KM has various roles, including the maintenance of sister chromatid cohesion or the recruitment of checkpoint proteins. KM binds to Bub1, which further recruits various factors required for the maintenance of sister chromatid cohesion, including the cohesin complex. The cohesin complex, H3T3ph, and CPC localize on entire chromosomes until prophase, but they localize to centromeric regions in metaphase cells. KM appears to play an essential role in proper centromere localization of H3T3ph or CPC in metaphase (Fig. 3f), since these localizations are dispersed into entire chromosomes in mutant cells lacking the Mis12C-CENP-T interaction. KM either directly or indirectly facilitates the removal of H3T3ph or CPC from chromosome arms and protects their removal from centromeres in metaphase for sister chromatid cohesion.

The functions of KM and N-N on the CENP-T pathway are relatively predictable based on previous observations[2,6], however, it is surprising to find that KM and N-N are separable, which means that Mis12C is not necessary for binding to Ndc80C. This striking result contrasts with a previous yeast study[62], which proposed that the microtubule-binding activity of yeast Ndc80C is regulated by its interaction with the MIND complex (yeast Mis12 complex).

For the CENP-C-KMN network interaction, Mis12C is the sole platform for Ndc80C; therefore, this interaction is critical. On the CENP-T-KMN network interaction, as one copy of Ndc80C is not sufficient, the Mis12C-Ndc80C interaction is critical in chicken cells. However, if Ndc80C is artificially supplied to CENP-T, this Ndc80C-recruitment function for Mis12C becomes dispensable (Fig. 5). Although this was observed in an artificial chicken situation, the Ndc80C-Mis12C interaction is dispensable in human cells, since humans have three copies of Ndc80C on CENP-T (KMN-N-N) (Fig. 7a)[35,41]. To bind to Ndc80C, CENP-T phosphorylation of the threonine residue of the TPR motif in its N-terminus is critical (Fig. 7c)[39,41]. Two TPR motifs exist in primate CENP-T, albeit a TPR motif in the extreme N-terminus of CENP-T is not clearly detected in CENP-Ts other than the primate CENP-T, while Chinese hamsters have two TPR motifs (Fig. 7c). This suggests that the TPR motif was acquired after the primate lineage to ensure robust kinetochore-microtubule interactions. Although we prefer to interpret that the copy number of Ndc80C binding sites on CENP-T is a critical factor for the increase in Ndc80C amounts in most organisms, it is also possible that the amount of Ndc80C would be increased in different ways. Therefore, this point is important and needs to be addressed.

While KM and N-N are separable, one copy of Ndc80C is usually supplied via Mis12C binding; however, the advantage of this system is unknown. This may be related to the Mis12C recruitment mechanism on the CENP-T pathway. Direct binding of Ndc80C to CENP-T facilitates the recruitment of Mis12C (Fig. 2), which further recruits additional copy of Ndc80C. This is an efficient way to recruit essential factors to the CENP-T pathway, and it is not necessary to have two Ndc80-binding sites in CENP-T. In in vitro experiments, Mis12C can bind to CENP-T, even if CENP-T does not directly bind to Ndc80C[41]. Therefore, the efficient Mis12C recruitment mechanism via direct Ndc80C binding to CENP-T cannot be explained only by performing in vitro studies. It would be an important further challenge to clarify the regulatory mechanisms of Mis12C recruitment onto the CENP-T pathway.

The kinetochore is a multi-protein complex with complicated protein-protein networks that often exhibit redundant functions. Therefore, it is necessary to dissect the functional roles of each kinetochore complex. In this study, using precise information on the binding sites of each complex, we defined the distinct functions of KM and N-N on the CENP-T pathway and demonstrated that they are separable. It is difficult to distinguish these functions using only a simple knockout

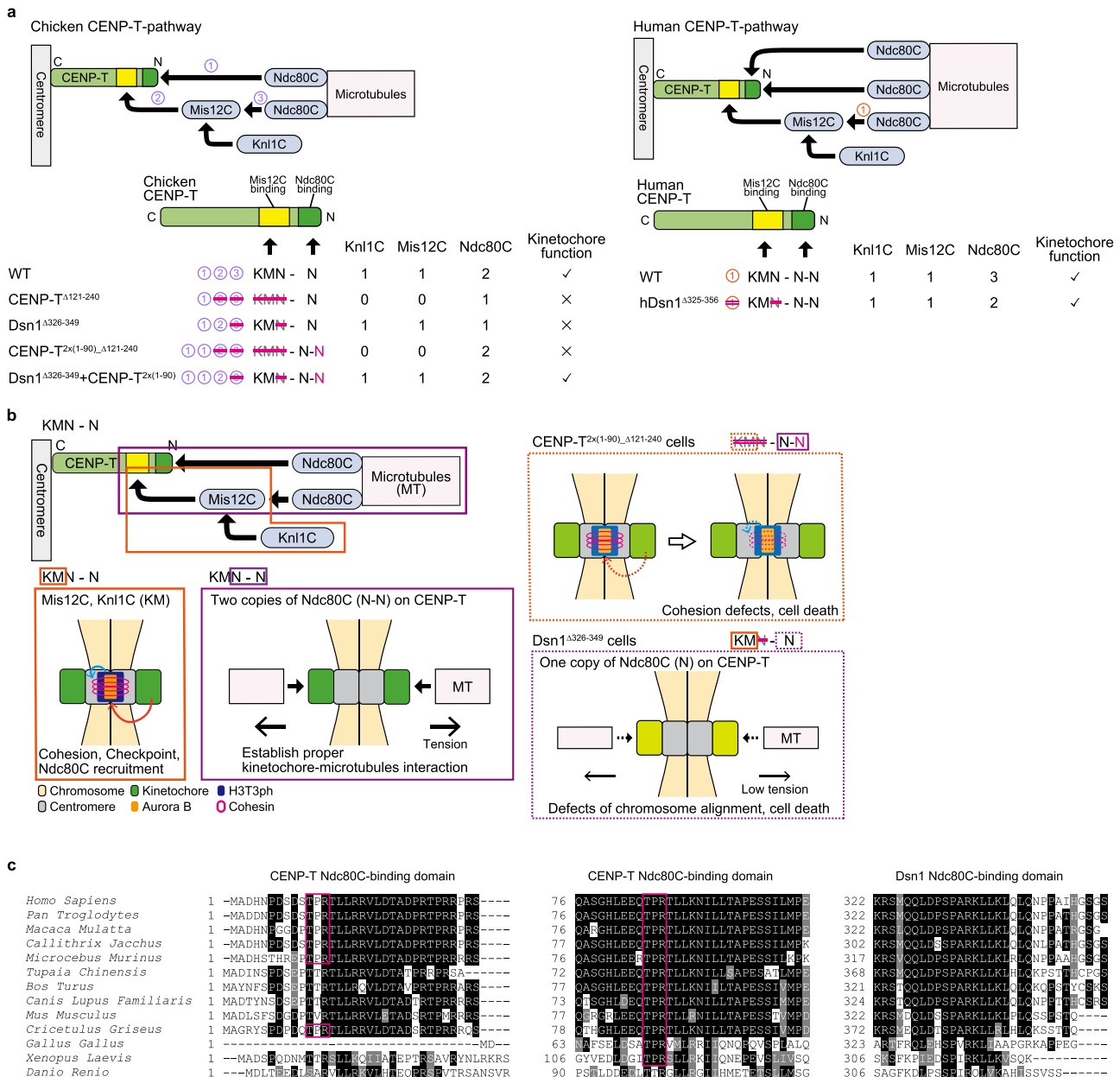

**Fig. 7 The KMN network plays distinct roles on CENP-T. a** Summary of numbers of K, M, and N in various mutants and their phenotypes in chicken and human cells. **b** A model of the essential roles of the KMN network on the CENP-T pathway. KM recruits Bub1, H3T3ph, and Aurora B to maintain sister chromatid cohesion and bind to checkpoint proteins. In addition, KM recruits one Ndc80C copy. However, two copies of Ndc80C (N-N) are critical for the establishment of kinetochore-microtubule interactions. As Ndc80C is not necessary to bind to Mis12C, KM and N-N are separable. In CENP-T$^{2X(1-90)\_\Delta121-240}$ cells (N-N lacking KM), cohesion defects were still observed, even if sufficient amounts of Ndc80C were bound to kinetochores. In Dsn1$^{\Delta326-349}$ cells (KM-N lacking one N), only one copy of Ndc80 localized kinetochores, and proper chromosome alignments did not occur, while cohesion defects were not observed. **c** Amino acid sequence alignments of the CENP-T extreme N-terminal region, conserved second N-terminal region, and Dsn1 C-terminal end for Ndc80C binding between various species.

approach for the components of the KMN network. Furthermore, functional kinetochores could be created by providing an artificial Ndc80C-binding site on CENP-T to explain the distinct KMN functions in the CENP-T pathway. Therefore, to control or manipulate kinetochore functions in the future, we must understand the precise roles of each complex in vivo, including redundant functions. We believe that this study provides a model for analyzing the precise function of each kinetochore complex in vivo and provides critical insights into kinetochore functions.

## Methods

**Cell culture**. Chicken DT40 cells[63] were cultured in Dulbecco's modified Eagle's medium (Nacalai tesque, 08459-64) containing 10% fetal bovine serum (Sigma, 172012-500 mL), 1% chicken serum (Thermo Fisher, 16110-082), 10 μM 2-Mercaptoethanol (Sigma, M3148), and penicillin (100 unit/mL)-streptomycin (100 μg/mL) (Thermo Fisher, 15140-122), at 38.5 °C with 5% CO$_2$. For gene inactivation in tetracycline-dependent conditional knockout cell lines (cKO-CENP-T, cKO-Dsn1, and cKO-Spc25 cells), 2 μg/ml tetracycline (Sigma, T-7660) was added. For degradation of GFP-AID-Nsl1 in cKO-Nsl1 (AID) cells, 500 μM 3-Insole acetic acid (IAA; Wako, 090-07123) was added. For examination of the cell growth for each cKO cell line, tetracycline or IAA was added to the culture medium at 0 h and the cell numbers were measured at each time point. Cell

numbers were normalized to that before tetracycline or IAA addition (0 h) for each cell line.

Human RPE-1 cells were cultured in Dulbecco's modified Eagle's medium containing 10% fetal bovine serum and penicillin (100 unit/mL)-streptomycin (100 μg/mL) at 37 °C with 5% $CO_2$. For degradation of GFP-AID-hDsn1 in cKO-hDsn1 (AID) cells, 250 μM IAA was added. For examination of the cell growth for each cKO cell line, IAA was added to the culture medium at 0 h and the cell numbers were measured at each time point. Cell numbers were normalized to that before IAA addition (0 h) for each cell line.

**Plasmid constructions for cell transfection.** Several CENP-T mutants (CENP-$T^{\Delta121-160}$, CENP-$T^{\Delta161-200}$, CENP-$T^{\Delta201-216}$, CENP-$T^{\Delta217-230}$, CENP-$T^{\Delta231-240}$, CENP-$T^{8A}$, CENP-$T^{2x(1-90)}$, and CENP-$T^{2x(1-90)\_\Delta121-240}$) were generated by PCR based mutagenesis or In-Fusion® HD Cloning Kit (Takara, 639648). To express 3X FLAG-fused CENP-T mutants, CENP-$T^{WT}$, CENP-$T^{\Delta90}$, CENP-$T^{T72A-S88A}$ [39], CENP-$T^{\Delta121-240}$[35], CENP-$T^{\Delta121-160}$, CENP-$T^{\Delta161-200}$, CENP-$T^{\Delta201-216}$, CENP-$T^{\Delta217-230}$, CENP-$T^{\Delta231-240}$, CENP-$T^{8A}$, CENP-$T^{2x(1-90)}$, and CENP-$T^{2x(1-90)\_\Delta121-240}$ were cloned into p3X FLAG-CMV-10 (Sigma, E7658).

To express 3X FLAG-fused CENP-T mutants under control of the endogenous *CENP-T* promoter in DT40 cells, 3X FLAG-fused CENP-T mutants and the Neomycin resistance gene expressed under the control of the *SV40* promoter (SV40 $Neo^R$) were cloned into the pBSKS_CT 2k vector, in which about 2 kb genome fragment (5′ and 3′ homology arm regions (approximately 1 kb each)) around the start codon of *CENP-T* is cloned[35] by In-Fusion® HD Cloning Kit (pBSKS_CT KI_3X FLAG-CENP-$T^{WT}$_SV40 $Neo^R$, pBSKS_CT KI_3X FLAG-CENP-$T^{\Delta90}$_SV40 $Neo^R$, pBSKS_ CT KI_3X FLAG-CENP-$T^{T72A-S88A}$_SV40 $Neo^R$, pBSKS_ CT KI_3X FLAG-CENP-$T^{\Delta121-240}$_SV40 $Neo^R$, pBSKS_CT KI_3X FLAG-CENP-$T^{\Delta121-160}$_SV40 $Neo^R$, pBSKS_ CT KI_3X FLAG-CENP-$T^{\Delta161-200}$_SV40 $Neo^R$, pBSKS_ CT KI_3X FLAG-CENP-$T^{\Delta201-216}$_SV40 $Neo^R$, pBSKS_ CT KI_3X FLAG-CENP-$T^{\Delta217-230}$_SV40 $Neo^R$, pBSKS_CT KI_3X FLAG-CENP-$T^{\Delta231-240}$_SV40 $Neo^R$, pBSKS_CT KI_3X FLAG-CENP-$T^{8A}$_SV40 $Neo^R$, pBSKS_ CT KI_3X FLAG-CENP-$T^{2x(1-90)}$_SV40 $Neo^R$, and pBSKS_CT KI_3X FLAG-CENP-$T^{2x(1-90)\_\Delta121-240}$_SV40 $Neo^R$). To disrupt the expression of the endogenous CENP-T, the Neomycin resistance cassette (SV40 $Neo^R$) was cloned into the pBSKS_CT 2k vector (pBSKS_ CT KI_SV40 $Neo^R$). To integrate these constructs into the endogenous *CENP-T* locus by using CRISPR/Cas9-mediated homologous recombination, the pX335_ggCENP-T containing single-guide RNA (sgRNA) against a genomic sequence around the start codon of *CENP-T* and SpCas9 nickase (D10A) gene[35] was used.

The mScarlet-fused CENP-$C^{\Delta73}$ expression vector was generated by cloning of CENP-$C^{\Delta73}$ cDNA into the pmScarlet-C1 (Addgene, 85042) vector. To express mScarlet-fused CENP-$C^{\Delta73}$ under control of the endogenous *β-actin* (*ACTB*) promoter in DT40 cells, mScarlet-fused CENP-$C^{\Delta73}$ and the Hygromycin B resistance gene expressed under control of the *PGK* promoter (PGK Hygro$^R$) were cloned into the pBSKS_ACTB 2k vector, in which about 2 kb genome fragment (5′ and 3′ homology arm regions (approximately 1 kb each)) around the *ACTB* start codon is cloned[35], by In-Fusion® HD Cloning Kit (pBSKS_ACTB KI_mScarlet-CENP-$C^{\Delta73}$_PGK Hygro$^R$). To express the HA-fused CENP-$C^{\Delta73}$ under control of the endogenous *ACTB* promoter in DT40 cells, mScarlet region of the pBSKS_ACTB KI_mScarlet-CENP-$C^{\Delta73}$_PGK Hygro$^R$ was replaced with the HA-tag (pBSKS_ACTB KI_HA-CENP-$C^{\Delta73}$_PGK Hygro$^R$). To integrate these constructs into the endogenous *ACTB* locus by using CRISPR/Cas9-mediated homologous recombination, pX335_ggACTB containing sgRNA against a genomic sequence around the start codon of *ACTB* and SpCas9 nickase (D10A) gene[35] was used.

To express the GFP-fused Dsn1$^{WT}$ or Dsn1$^{\Delta326-349}$ [35], under control of the *Dsn1* promoter in DT40 cells, the GFP-fused Dsn1$^{WT}$ or Dsn1$^{\Delta326-349}$ with puromycin resistance gene expressed by the *PGK* promoter (PGK Puro$^R$), EcoGPT expressed by the *PGK* promoter (PGK EcoGPT), or the blasticidin S resistance gene expressed by *ACTB* promoter (ACTB BSR) were cloned into the pBSKS_Dsn1 2k vector, in which about 2 kb genome fragment (5′ and 3′ homology arm regions (approximately 1 kb each)) around the *Dsn1* start codon is cloned, by In-Fusion® HD Cloning Kit (pBSKS_Dsn1 KI_GFP-Dsn1$^{WT}$_PGK Puro$^R$, pBSKS_Dsn1 KI_GFP-Dsn1$^{WT}$_PGK EcoGPT, pBSKS_Dsn1 KI_GFP-Dsn$^{WT}$_ACTB BSR, and pBSKS_Dsn1 KI_GFP-Dsn1$^{\Delta326-349}$_ACTB BSR). To integrate these constructs into the endogenous *Dsn1* locus by using CRISPR/Cas9-mediated homologous recombination, we designed sgRNA against a DNA sequence (GCGCGCGTAGCCGCCATGGA) around the start codon of *Dsn1* using CRISPOR[64], and cloned it into the BbsI site of the pX330 plasmid (Addgene, 42230) containing SpCas9 gene (pX330_ggDsn1).

To express the GFP-fused Nuf2 under control of the *Nuf2* promoter in DT40 cells, GFP-fused Nuf2 and PGK Puro$^R$ or PGK EcoGPT were cloned into pBSKS_Nuf2 2k vector, in which about 2 kb genome fragment (5′ and 3′ homology arm regions (approximately 1 kb each)) around the end of the exon1 of Nuf2 genomic sequence is cloned, by In-Fusion® HD Cloning Kit (pBSKS_Nuf2 KI_Nuf2-GFP_PGK Puro$^R$ and pBSKS_Nuf2 KI_Nuf2-GFP_PGK EcoGPT). To express the mScarlet-fused Nuf2 under control of the *Nuf2* promoter in DT40 cells, mScarlet-fused Nuf2 and ACTB BSR or PGK EcoGPT were cloned into pBSKS_Nuf2 2k vector by In-Fusion® HD Cloning Kit

(pBSKS_Nuf2 KI_Nuf2-mScarlet_ACTB BSR and pBSKS_Nuf2 KI_Nuf2-mScarlet_PGK EcoGPT). To integrate these constructs into the endogenous *Nuf2* locus by using CRISPR/Cas9-mediated homologous recombination, we designed sgRNA against a DNA sequence (GCCACCCTTAAGGTGTGTG) in the intron 1 of the *Nuf2* genomic sequence using CRISPOR[64], and cloned it into the BbsI site of the pX330 (pX330_ggNuf2).

To disrupt expression from the endogenous *CENP-C* gene by using CRISPR/Cas9 system, sgRNA against a DNA sequence in the exon2 of the *CENP-C* genomic sequence (CGAGCAAGATTCTGCGGGCA) was designed using CRISPOR[64], and cloned it into the BbsI site of the pX330 (pX330_ggCENP-C).

To express the GFP-fused CENP-$T^{2x(1-90)}$, pEGFP-C3_CENP-$T^{2x(1-90)}$ was generated from the pEGFP-C3_CENP-$T^{WT}$ by In-Fusion® HD Cloning Kit.

To generate cKO-Nsl1 (AID) cells, we used CRISPR/Cas9-meidated genome editing combined with the auxin degron-based cKO system[65–67]. To disrupt the expression of the endogenous *Nsl1* by using CRISPR/Cas9 system, sgRNA against a DNA sequence (TAAGAAATCATTACCTTGAG) across the boundary between the exon 4 and the intron 4 of the *Nsl1* gene was designed using CRISPOR[64], and cloned it into the Bbs1 site of pX330 (pX330_ggNsl1). To express the OsTIR1-T2A-BSR and GFP-mAID-fused Nsl1 simultaneously under control of the *CMV* promoter, Nsl1 cDNA was cloned into the pAID1.2-NEGFP containing CMV promoter_OsTIR1-T2A-BSR_IRES_GFP-mAID (pAID1.2-NEGFP-Nsl1). To linearize the pAID1.2-NEGFP-Nsl1, the pX330_pAID1.2 containing sgRNA against a back bone sequence of the pAID1.2-NEGFP-Nsl1 and SpCas9 gene was used[67].

To express the GFP-fused Nsl1$^{WT}$, the pEGFP-C3_Nsl1$^{WT}$ was used, and leucine 220 and leucine 222 residues of pEGFP-C3_Nsl1$^{WT}$ were substituted with glutamic acid (pEGFP-C3_Nsl1$^{EE}$) by PCR based mutagenesis for the expression of GFP-fused Nsl1$^{EE}$.

To express the VH-mScarlet under control of the endogenous *ACTB* promoter in DT40 cells for the tension sensor assay, GFP-sequence of VH-GFP (a gift from T. Maresca, University of Massachusetts Amherst, Amherst, MA, USA) was replaced with mScarlet (VH-mScarlet). Then, VH-mScarlet and PGK EcoGPT were cloned into pBSKS_ACTB 2k vector (pBSKS_ACTB 2k_VH-mScarlet).

To express the OsTIR1-T2A-BSR and GFP-mAID-fused hDsn1 simultaneously under control of the *CMV* promoter, hDsn1 cDNA was cloned into the pAID1.2-NEGFP (pAID1.2-NEGFP-hDsn1). To express the OsTIR1-T2A-BSR and GFP-mAID-hDsn1 under control of the *CMV* promoter from the *AAVS1* locus in RPE-1 cells, CMV promoter_OsTIR1-T2A-BSR_IRES_GFP-mAID-hDsn1 fragment was cloned into the pBSKS_AAVS1 1.6k vector, in which about 1.6 kb genome fragment (5′ and 3′ homology arm regions (approximately 0.8 kb each)) around the AAVS1 intron 1 is cloned, by In-Fusion® HD Cloning Kit. To integrate this construct into the endogenous *AAVS1* locus by using CRISPR/Cas9-mediated homologous recombination, we designed sgRNA against a DNA sequence (TAGTGGCCCCACTGTGGGGT) in the intron 1 of *AAVS1* using CRISPOR[64], and cloned it into the Bbs1 site of the pX330 (pX330_hsAAVS1).

To express the mScarlet-fused hDsn1 expressed under control of the endogenous *hDsn1* promoter in RPE-1 cells, mScarlet-fused hDsn1$^{WT}$ or hDsn1$^{\Delta325-356}$ and Neo$^R$ or Puro$^R$ expressed by *ACTB* promoter (ACTB Neo$^R$, ACTB Puro$^R$) were cloned into pBSKS_hDsn1 2k vector, in which about 2 kb genome fragment (5′ and 3′ homology arm regions (approximately 1 kb each)) around the *hDsn1* start codon is cloned, by In-Fusion® HD Cloning Kit (pBSKS_hDsn1 KI_mScarlet-hDsn1$^{WT}$_ACTB Neo$^R$, pBSKS_hDsn1 KI_mScarlet-hDsn1$^{WT}$_ACTB Puro$^R$, pBSKS_hDsn1 KI_mScarlet-hDsn1$^{\Delta325-356}$_ACTB Neo$^R$, and pBSKS_hDsn1 KI_mScarlet-hDsn1$^{\Delta325-356}$_ACTB Puro$^R$). To disrupt the expression of endogenous hDsn1 in RPE-1 cells, ACTB Neo$^R$ or ACTB Puro$^R$ were cloned into pBSKS_hDsn1 2k vector by In-Fusion® HD Cloning Kit (pBSKS_hDsn1 KI_ ACTB Neo$^R$, pBSKS_hDsn1 KI_ACTB Puro$^R$). To integrate these constructs into the endogenous *hDsn1* locus by using CRISPR/Cas9-mediated homologous recombination, we designed sgRNA against a DNA sequence (CTTACCTTGGGGTTCAGGCTT) in the intron1 of the hDsn1 genomic sequence using CRISPOR[64], and cloned it into the Bbs1 site of pX330 (pX330_hsDsn1).

**Generation of cell lines.** The cell lines established in this study are listed in Supplementary Table 1.

To express the mScarlet-fused CENP-$C^{\Delta73}$ or HA-fused CENP-$C^{\Delta73}$ under control of the endogenous *ACTB* promoter, pBSKS_ACTB KI_mScarlet-CENP-$C^{\Delta73}$_PGK Hygro$^R$ or pBSKS_ACTB KI_HA-CENP-$C^{\Delta73}$_PGK Hygro$^R$ were transfected with pX335_ggACTB using Neon Transfection System (Thermo Fisher, MPK5000) with 6 times of pulse (1400 V, 5 msec) in Tet-Off CENP-T/CENP-$T^{\Delta/+}$ cells[5] (mScarlet-CENP-$C^{\Delta73}$), cKO-Spc25 cells[39] (mScarlet-CENP-$C^{\Delta73}$), and CL18 cells (HA-CENP-$C^{\Delta73}$). The transfected cells were selected in the medium containing 2.5 mg/mL Hygromycin B (Wako, 087-06152) in 96 well plates to isolate single-cell clones.

To knockout the expression of endogenous *CENP-C*, pX330_ggCENP-C was transfected with pBSKS_ACTB-BSR using Neon Transfection System with six times of pulse (1400 V, 5 msec) in Tet-OFF CENP-T/CENP-$T^{\Delta/+}$ expressing mScarlet-CENP-$C^{\Delta73}$ cells, cKO-Spc25 cells expressing mScarlet-CENP-$C^{\Delta73}$, and CL18 cells expressing HA-CENP-$C^{\Delta73}$. The transfected cells were selected in the medium containing 25 μg/mL Blasticidin S hydrochloride (Wako, 029-18701) for

48 h, and seeded in 96 well plates to isolate single-cell clones (Tet-Off CENP-T/CENP-T$^{\Delta/+}$/CENP-C$^{\Delta73}$ cells, cKO-Spc25-CC$^{\Delta73}$ cells, and CENP-C$^{\Delta73}$ cells).

To express 3X FLAG-fused CENP-T constructs under the control of the endogenous *CENP-T* promoter, each 3X FLAG-fused CENP-T construct (pBSKS_CT KI_3X FLAG-CENP-T$^{mutant}$_SV40 Neo$^R$) was transfected with pX335_ggCENP-T using Neon Transfection System with 6 times of pulse (1400 V, 5 msec) in Tet-Off CENP-T/CENP-T$^{\Delta/+}$ cells or Tet-Off CENP-T/CENP-T$^{\Delta/+}$/CENP-C$^{\Delta73}$ cells. The transfected cells were selected in the medium containing 2 mg/mL G418 (Santa Cruz Biotechnology, SC-29065B) in 96 well plates to isolate single-cell clones (cKO-CT-CC$^{WT}$/CENP-T$^{mutant}$ cells, cKO-CT-CC$^{\Delta73}$/CENP-T$^{mutant}$ cells).

To express the GFP-fused Dsn1 under control of the endogenous *Dsn1* promoter, pBSKS_Dsn1 KI_GFP-Dsn1$^{WT}$_PGK Puro$^R$ and pBSKS_Dsn1 KI_GFP-Dsn1$^{WT}$_PGK EcoGPT were transfected with pX330_ggDsn1 using Neon Transfection System with 6 times of pulse (1400 V, 5 msec) in cKO-CT-CC$^{\Delta73}$ cells, cKO-CT-CC$^{\Delta73}$/CENP-T$^{WT}$ cells, cKO-CT-CC$^{\Delta73}$/CENP-T$^{\Delta90}$ cells, cKO-CT-CC$^{\Delta73}$/CENP-T$^{\Delta121-240}$ cells, cKO-CT-CC$^{\Delta73}$/CENP-T$^{\Delta121-160}$ cells, cKO-CT-CC$^{\Delta73}$/CENP-T$^{\Delta161-200}$ cells, cKO-CT-CC$^{\Delta73}$/CENP-T$^{\Delta201-216}$ cells, cKO-CT-CC$^{\Delta73}$/CENP-T$^{\Delta217-230}$ cells, cKO-CT-CC$^{\Delta73}$/CENP-T$^{\Delta231-240}$ cells, cKO-CT-CC$^{\Delta73}$/CENP-T$^{8A}$ cells, cKO-Spc25-CC$^{\Delta73}$ cells, and CENP-C$^{\Delta73}$ cells. The transfected cells were selected in the medium containing 0.5 μg/mL puromycin (Takara, Z1305N), 25 μg/mL mycophenolic acid (Wako, 138-11003), and 125 μg/mL xanthine (Sigma, 1002581797) in 96 well plates to isolate single-cell clones. pBSKS_Dsn1 KI_GFP-Dsn1$^{WT}$_ACTB BSR or pBSKS_Dsn1 KI_GFP-Dsn1$^{\Delta326-349}$_ ACTB BSR transfected with the pX330_ggDsn1 using Neon Transfection System with 6 times of pulse (1400 V, 5 msec) in Tet-Off Dsn1/Dsn1$^{\Delta/+}$ cells, and then the transfected cells were selected in the medium containing 25 μg/mL Blasticidin S hydrochloride in 96 well plates to isolate single-cell clones (cKO-Dsn1/Dsn1$^{WT}$ cells and cKO-Dsn1/Dsn1$^{\Delta326-349}$ cells).

To express the GFP-fused Nuf2 under control of the endogenous *Nuf2* promoter, pBSKS_Nuf2 KI_GFP-Nuf2_PGK Puro$^R$ and pBSKS_Nuf2 KI_GFP-Nuf2_PGK EcoGPT were transfected with the pX330_ggNuf2 using Neon Transfection System with 6 times of pulse (1400 V, 5 msec) in cKO-CT-CC$^{\Delta73}$ cells, cKO-CT-CC$^{\Delta73}$/CENP-T$^{WT}$ cells, cKO-CT-CC$^{\Delta73}$/CENP-T$^{\Delta90}$ cells, cKO-CT-CC$^{\Delta73}$/CENP-T$^{T72A-S88A}$ cells, cKO-CT-CC$^{\Delta73}$/CENP-T$^{\Delta121-240}$ cells, cKO-CT-CC$^{\Delta73}$/CENP-T$^{\Delta161-200}$ cells, cKO-CT-CC$^{\Delta73}$/CENP-T$^{\Delta201-216}$ cells, cKO-CT-CC$^{\Delta73}$/CENP-T$^{2x(1-90)}$ cells, and CENP-C$^{\Delta73}$ cells. The transfected cells were selected in the medium containing 0.5 μg/mL puromycin, 25 μg/mL mycophenolic acid, and 125 μg/mL xanthine in 96 well plates to isolate single-cell clones.

To express the mScarlet-fused Nuf2 under control of the endogenous *Nuf2* promoter, pBSKS_Nuf2 KI_Nuf2-mScarlet_ACTB BSR and pBSKS_Nuf2 KI_Nuf2-mScarlet_PGK EcoGPT were transfected with the pX330_ggNuf2 using Neon Transfection System with 6 times of pulse (1400 V, 5 msec) in cKO-Dsn1/GFP-Dsn1$^{WT}$ cells and cKO-Dsn1/GFP-Dsn1$^{\Delta326-349}$ cells. The transfected cells were selected in the medium containing 25 μg/mL Blasticidin S hydrochloride in 96 well plates to isolate single-cell clones.

To generate cKO-Nsl1 (AID) cells, the pAID1.2-NEGFP-Nsl1 was transfected with the pX330_ggNsl1 and pX330_pAID1.2 using Neon Transfection System with 6 times of pulse (1400 V, 5 msec) in CL18 cells. The transfected cells were selected in the medium containing 25 μg/mL Blasticidin S hydrochloride in 96 well plates to isolate single-cell clones.

To express the GFP-fused Nsl1$^{WT}$ and Nsl1$^{EE}$, the linearized pEGFP-C3_Nsl1$^{WT}$ or pEGFP-C3_Nsl1$^{EE}$ was transfected using Gene Pulser II electroporator (Bio-Rad, 165-2105) in cKO-Nsl1 (AID) cells. The transfected cells were selected in the medium containing 2 mg/mL G418 in 96 well plates to isolate single-cell clones.

To express the GFP-fused CENP-T$^{2x(1-90)}$, the linearized pEGFP-C3_CENP-T$^{2x(1-90)}$ was transfected using Gene Pulser II electroporator in cKO-Dsn1/GFP-Dsn1$^{\Delta326-349}$ cells. The transfected cells were selected in the medium containing 2 mg/mL G418 in 96 well plates to isolate single-cell clones.

For the tension sensor assay, pBSKS_CT 2k_TagRFP-CENP-T-TR (241-529;TR)[35] was transfected with the pX335_ggCT using Neon Transfection System with 6 times of pulse (1400 V, 5 msec) in cKO-Dsn1/Dsn1$^{WT}$ or cKO-Dsn1/Dsn1$^{\Delta326-349}$ cells. The transfected cells were selected in the medium containing 2 mg/mL G418 in 96 well plates to isolate single-cell clones. To express the VH-mScarlet, the pBSKS_ACTB 2k_VH-mScarlet was transfected with the pX335_ggACTB using Neon Transfection System with 6 times of pulse (1400 V, 5 msec) in cKO-Dsn1/Dsn1$^{WT}$ or cKO-Dsn1/Dsn1$^{\Delta326-349}$ cells expressing the TagRFP-CENP-T-TR. The transfected cells were selected in the medium containing 25 μg/mL mycophenolic acid and 125 μg/mL xanthine in 96 well plates to isolate single-cell clones.

To express the OsTIR1-T2A-BSR and GFP-mAID-fused hDsn1, pAID1.2-NEGFP-hDsn1 was transfected with the pX330_hsAAVS1 using Neon Transfection System with 6 times of pulse (1400 V, 5 msec) in RPE-1 cells. The transfected cells were selected in the medium containing 10 μg/mL Blasticidin S hydrochloride. To disrupt the expression of endogenous hDsn1 in RPE-1 cells expressing the OsTIR1-T2A-BSR and GFP-mAID-hDsn1, pBSKS_hDsn1 KI_ACTB Neo$^R$ and pBSKS_hDsn1 KI_ACTB Puro$^R$ were transfected with the pX330_hDsn1 using Neon Transfection System with 6 times of pulse (1400 V, 5 msec). To replace endogenous hDsn1 with mScarlet-hDsn1$^{WT}$ or mScarlet-hDsn1$^{\Delta325-356}$ in RPE1 cells expressing OsTIR1-T2A-BSR and GFP-mAID-hDsn1, the pBSKS_hDsn1 KI_mScarlet-hDsn1$^{WT}$_ACTB Neo$^R$ and

pBSKS_hDsn1 KI_ mScarlet-hDsn1$^{WT}$_ACTB Puro$^R$ or pBSKS_hDsn1 KI_mScarlet-hDsn1$^{\Delta325-356}$_ACTB Neo$^R$ and pBSKS_hDsn1 KI_ mScarlet-hDsn1$^{\Delta325-356}$_ACTB Puro$^R$ were transfected with the pX330_hDsn1 using Neon Transfection System with 6 times of pulse (1400 V, 5 msec). The transfected cells were selected in the medium containing 0.5 mg/ml G418 and 2 μg/ml puromycin.

**Genotyping.** For the genomic DNA extraction from DT40 cells, cells were collected, washed with PBS (Nissui Pharmaceutical Co., Ltd., 05913), and suspended in 0.2 mg/mL Proteinase K (Sigma, P2308) in 0.1% PBST (PBS, 0.1% Tween 20 (Nacalai tesque, 28353-85)). The suspended cells were incubated at 55 °C for 90 min and heated at 96 °C for 15 min.

For the genomic DNA extraction of RPE-1 cells, cells were collected after trypsinizing, washed with PBS, and suspended in 0.2 mg/mL Proteinase K in 0.1% PBST (PBS, 0.1% Tween 20). The suspended cells were incubated at 55 °C for 90 min and heated at 96 °C for 15 min.

To investigate the mutation sequence in the chicken *CENP-C gene*, the genomic regions flanking the sgRNA target site were amplified by PCR using the following primers: Fw: 5'- ATGCATCAACCAGGAGGCTGTC -3', Rv: 5'- CTAAGGCACA CCATTAGTTTTGG -3'. The PCR amplicons were cloned into pBluescript II SK (-) and sequenced.

The target integration of several constructs was confirmed by PCR. The primers used for PCRs were listed in Supplementary Table 2.

**Immunoblot.** DT40 cells were collected, washed with cold PBS, and suspended in 1xLaemmli Sample Buffer (LSB; 62.5 mM Tris (Sigma, T1503) –HCl (Nacalai tesque, 18321-05) (pH 6.8), 10% Glycerol (Nacalai tesque, 17018-83), 2% SDS (Nacalai tesque, 02873-75), 5% 2-mercaptoethanol (Sigma, M3148), bromophenol blue (Wako, 101123)) (final concentration $1 \times 10^4$ cells/μL). Following the sonication, the lysate was heated at 96 °C for 5 min.

RPE-1 cells were collected after trypsinization, washed with cold PBS, and suspended in 1xLSB (final concentration $1 \times 10^4$ cells/μL). Following the sonication, the lysate was heated at 96 °C for 5 min.

The collected samples ($5 \times 10^4$ cells) were separated by SDS-PAGE (SuperSep Ace, 5–20% (Wako, 194-15021) or handmade 7.5%) and transferred onto PVDF membranes (Immobilon®-P (Merck, IPVH00010)). The membrane was probed with primary antibody diluted with Signal Enhancer Hikari (Nacalai Tesque, 02270-81) at 4 °C for overnight. After washing with 0.1% TBST (TBS: 50 mM Tris-HCl, 150 mM NaCl; 0.1% Tween 20) for 15 min, the membrane was probed with secondary antibody diluted with Signal Enhancer Hikari at room temperature for 1 h. After washing with 0.1% TBST for 15 min, the signals were detected using ECL Prime Western Blotting Detection Reagent (cytiva, RPN2232) and visualized by ChemiDoc Touch imaging system (Bio-Rad). The Image processing was performed using Image Lab 5.2.1 (Bio-Rad) and Adobe Photoshop v23.1.0 (Adobe).

Primary antibodies used in immunoblot analyses were rabbit anti-chicken CENP-T[5] (1:10000), rabbit anti-chicken CENP-C[18] (in this study) (1:10000), rabbit anti-chicken Dsn1[35] (1:10000), rabbit anti-chicken Nuf2[68] (1:5000), rabbit anti-chicken Ndc80[68] (1:2000), rabbit anti-chicken Spc25 (in this study using recombinant chicken full-length Spc25 as an immunogen) (1:2000), rabbit anti-chicken Knl1[34] (1:2000), rabbit anti-chicken Mis12[42] (1:2000), rabbit anti-human Dsn1 (Bio Academia, 70-101) (1:5000), rabbit anti-GFP (MBL, 598) (1:2000), mouse anti-FLAG M2 (Sigma, F1804) (1:1000), rat anti-RFP (Chromotek, 5f8) (1:2000), and mouse anti-α-tubulin (Sigma, T9026) (1:5000). Secondary antibodies used in immunoblot analysis were horseradish peroxidase-conjugated (HRP)-conjugated goat anti-rabbit IgG (Jackson ImmunoResearch, 111-035-144) (1:10000), HRP-conjugated goat anti-rat IgG (Jackson ImmunoResearch, 112-035-003) (1:10000), and HRP-conjugated rabbit anti-mouse IgG (Jackson ImmunoResearch, 115-035-003) (1:10000).

**Immunoprecipitation.** For GFP-Dsn1 immunoprecipitation, cKO-Dsn1/Dsn1$^{WT}$ or cKO-Dsn1/Dsn1$^{\Delta326-349}$ cells were cultured with 2 μg/mL tetracycline for 36 h. For cKO-Dsn1/GFP-Dsn1$^{WT}$ cells cultured with 2 μg/mL tetracycline for 36 h, 100 ng/mL nocodazole (Sigma, M1404) were also added for last 10 h. For mScarlet-hDsn1 immunoprecipitation, cKO-Dsn1/hDsn1$^{WT}$ or cKO-Dsn1/hDsn1$^{\Delta325-356}$ cells were cultured with 100 ng/mL nocodazole for 24 h and treated with IAA for last 2 h. These cells were collected, washed with cold PBS twice, suspended in Lysis buffer (50 mM NaPi (Disodium Hydrogenphosphate, 197-02865; Sodium Dihydrogenphosphate Dihydrate, 192-02815) (pH 8.0), 300 mM NaCl (Nacalai tesque, 31320-05), 0.1% NP40 (Nacalai tesque, 25223-004), 5 mM 2-mercaptoethanol, 1xcOmplete EDTA-free proteinase inhibitor (Roche, 11873580001), and 1xphosphatase inhibitor (10 mM Sodium Pyrophosphate (Sigma, 221368), 5 mM Sodium Azide (Wako, 195-11092), 10 mM NaF (Wako, 192-01972), 0.4 mM Sodium orthovanadate (Sigma, S6508), 20 mM β-Glycerophosphate (Wako, 048-34332))) (final: $1 \times 10^8$ cells/mL for DT40 cells, $2 \times 10^7$ cells/mL for RPE1 cells) and sonicated. The lysate was clarified by centrifugation and the supernatant was incubated with GFP-Trap Magnetic agarose (Chromotek, gtma) or RFP-Trap Magnetic agarose (Chromotek, rtma) at 4 °C for 1 h. Proteins precipitated with GFP-Trap Magnetic agarose or RFP-Trap Magnetic agarose were washed with Lysis buffer

five times, eluted with 2 × LSB. The eluted samples were heated at 96 °C for 5 min and subjected to immunoblot analysis.

**Immunofluorescence analysis**. The cells were cytospan onto slide glasses or coverslips by the Cytospin3 centrifuge (Shandon, TH-CYTO3), fixed with 3% paraformaldehyde (PFA; Nacalai tesque, 26126-25) in 250 mM HEPES (Nacalai tesque, 17514-15) -NaOH (Nacalai tesque, 31511-05) (pH 7.4) at RT for 15 min, and permeabilized in 0.5% NP-40 in PBS at RT for 10 min. For the γ-tubulin staining, the cells were cultured on the Concanavalin A (Sigma, L7647) coated chamber slides for 30 min to promote cell attachment and fixed with 3% PFA in 250 mM HEPES-NaOH (pH 7.4) at RT for 10 s and Methanol at –25 °C for 10 min. After blocking with 0.5% BSA (Equitech-Bio Inc, BAC62) in PBS for 5 min, the cells were incubated with primary antibodies diluted in 0.5% BSA in PBS at 37 °C for 1 h. The cells were washed three times with 0.5% BSA in PBS, incubated with secondary antibodies diluted in 0.5% BSA in PBS at 37 °C for 1 h, and washed three times with 0.5% BSA in PBS. The cells were post-fixed for 10 min. DNA was stained with 100 ng/mL DAPI in PBS for 20 min. The stained cells were mounted with VECTASHIELD Mounting Medium (Vector Laboratories, H-1000).

Primary antibodies used in immunofluorescence analysis were rabbit anti-chicken Knl1[34] (1:5000), rabbit anti-chicken CENP-T[5] (1:2000), rabbit anti-chicken Bub1[34] (1:1000), rabbit anti-chicken Aurora B[34] (1:5000), mouse anti-H3T3ph[34] (1:10000), mouse anti-human Ndc80 (Abcam, ab3613) (1:1000), mouse anti-FLAG-M2 (Sigma, F1804) (1:1000), mouse anti-α-tubulin (Sigma T9026) (1:5000), rabbit anti-γ-tubulin (Sigma, T5192) (1:1000), FITC-conjugated mouse anti-α-tubulin (Sigma, F2168) (1:1000), mouse anti-human CENP-A[69] (1:200), and rat anti-human CENP-T (immunogen: HNPDSDSTPRTLLRRVLDTC, a gift from Kinya Yoda, Nagoya University, Nagoya, Japan) (1:1000). Secondary antibodies used in immunofluorescence analysis were FITC-conjugated goat anti-rabbit IgG F(ab')2 (1:1000), FITC-conjugated goat anti-mouse IgG (Jackson ImmunoResearch, 115-095-003) (1:1000), FITC-conjugated goat anti-rat IgG (Jackson ImmunoResearch, 112-095-003) (1:1000), Cy3-conjugated mouse anti-rabbit IgG (Jackson ImmunoResearch, 211-165-109) (1:1000), Cy3-conjugated goat anti-mouse IgG (Jackson ImmunoResearch, 115-165-003) (1:1000), and Alexa647-conjugated goat anti-mouse IgG (Jackson ImmunoResearch, 115-605-003) (1:1000).

For counterstaining of DNA, 100 ng/mL DAPI (Roche, 10236276001) in PBS was used for 20 min incubation. The stained cells were mounted with VECTASHIELD Mounting Medium.

**Chicken CENP-C antibody generation**. The chicken CENP-C fragment (amino acids 1-302) was cloned into pET30b (Merk, 69910). The His-tag fused chicken CENP-C (1-302) protein was expressed in an *Escherichia coli* cell line, Rosetta2 (DE3) (Merk, C2527I). Cells expressing His-tag fused chicken CENP-C (1-302) were harvested by centrifugation, resuspended in binding buffer (20 mM Tris-HCl (pH 7.5), 500 mM NaCl, 5 mM Imidazole (Wako, 095-00015), and 1×complete EDTA-free proteinase inhibitor), and lysed by sonication. The lysate was clarified by centrifugation and the supernatant was incubated with Ni-NTA beads (Qiagen, 30210). After washing Ni-NTA beads with wash buffer (20 mM Tris-HCl (pH 7.5), 500 mM NaCl, and 40 mM Imidazole), bound proteins were eluted with elute buffer (20 mM Tris-HCl (pH 7.5), 500 mM NaCl, and 800 mM Imidazole). The elution was dialyzed against PBS, concentrated using Amicon-Ultra-15 10 K (Merk, UFC901008), frozen in liquid nitrogen and stored at −80 °C. The purified protein was sent to Cosmo bio to raise antibodies against chicken CENP-C in rabbits.

**Chromosome spread**. For the preparation of nocodazole treated samples, the cells were cultured with 100 ng/mL nocodazole for 14 h or 500 ng/mL nocodazole for 4 h. For the preparation of MG132 (Sigma, C2211) treated samples, the cells with cultured with 100 ng/mL nocodazole for 10 h, washed with 1xPBS three times, and treated with 10 µM MG132 for 4 h. Nocodazole or MG132 treated cells were collected and suspended in 0.56% hypotonic buffer (40 mM KCl (Nacalai tesque, 28514-75), 10 mM HEPES/NaOH (pH 7.4), 0.5 mM EDTA (Nacalai tesque, 15111-45)) (Fig. 2h, 2i, 3c, 3d, 3e, 5h, Supplementary Fig. 5e, 5f, 5g). The cells (3.5–4 × 10^5 cells) were cytospan onto slide glasses or coverslips by the Cytospin3 centrifuge, and fixed with 3% PFA in 250 mM HEPES-NaOH (pH 7.4) at RT for 15 min. The samples were subjected to immunofluorescence analysis.

**Microtubule staining**. For the microtubule staining of DT40 cells, the cells were cultured on the Concanavalin A coated chamber slides for 30 min to promote cell attachment. Then, the cells were incubated with ice-cold medium for 10 min. After fixed with 3% PFA in 250 mM HEPES-NaOH (pH 7.4) at RT for 15 min, the samples were subjected to immunofluorescence analysis with an anti-α-tubulin antibody.

For the microtubule staining of RPE1 cells, the cells were cultured on the coverslips and incubated with the ice-cold medium for 10 min. After extracted with 0.2% triton X-100 (Sigma, T8787) in PHEM buffer (60 mM PIPES (Sigma, P1851), 25 mM HEPES, 10 mM EGTA (Sigma, E3889), and 2 mM MgCl₂ (Nacalai tesque, 20909-55), pH 7.4) for 1 min, equal amount of 4% PFA in PHEM was added (final 2% PFA) and fixed for 5 min. After removed the solution, the cells were fixed with 4% PFA in PHEM for 5 min, washed with 0.1% Triton X-100 in PBS (PBSTX) for

5 min, blocked with 3% BSA in PBSTX for 5 min, and incubated with primary antibodies diluted in 3% BSA in PBSTX at 37 °C for 1 h. The cells were washed three times with PBSTX, incubated with secondary antibodies diluted with 3% BSA in PBSTX at 37 °C for 1 h, and washed three times with PBSTX. DNA was stained with 100 ng/mL DAPI in PBS for 20 min. The stained cells were mounted with VECTASHIELD Mounting Medium.

**Image acquisition**. Immunofluorescence images and DAPI staining images were acquired every 0.2 µm intervals of z-slice using a Zyla 4.2 sCMOS camera (Andor) mounted on a Nikon Eclipse Ti inverted microscope with an objective lens (Nikon; Plan Apo lambda 100×/1.45 NA) with a spinning disk confocal scanner unit (CSU-W1; Yokogawa) controlled with NIS-elements v4.60 (Nikon). The images in the Figures are the maximum intensity projection or sum intensity projection (chromosome spread samples) of the Z-stack generated with Fiji (v1.53)[70] and were processed using Fiji (v1.53) and Adobe Photoshop v23.1.0 (Adobe).

**Evaluation of chromosome alignment in DT40 cells**. The cells were cultured on the Concanavalin A-coated chamber slides for 30 min to promote cell attachment. After fixed with 3% PFA in 250 mM HEPES-NaOH (pH 7.4) at RT for 15 min, the cells were subjected to immunofluorescence analysis. For the calculation of chromosome alignment value, positions of each kinetochore signal were plotted and linear approximation straight-line was drawn. Absolute values of a correlation coefficient were defined as a chromosome alignment value.

**Flow cytometry analysis**. The cells were treated with tetracycline (2 µg/mL) for 30 h and incubated with 20 µM BrdU (5-bromo-2-deoxyuridine; Sigma, 10280879001) for 20 min and harvested. The harvested cells were washed with the ice-cold PBS, fixed with ice-cold 70% ethanol and stored at −20 °C. The fixed cells were washed with 1% BSA/PBS, incubated in 4 N HCl with 0.5% Triton X-100 at RT for 30 min, and washed with 1% BSA/PBS three times. The cells were incubated with an anti-BrdU antibody (BD, 347580) at RT for 1 h and washed with 1% BSA/PBS three times. Then, the cells were incubated with FITC-conjugated antimouse IgG (Jackson ImmunoResearch, 115-095-003; at 1:20 dilution in 1% BSA/PBS) at RT for 30 min, washed with 1% BSA/PBS and stained DNA with propidium iodide (Sigma, P4170) (10 µg/mL) in 1% BSA/PBS at 4 °C overnight or RT for 4 h. The stained cells were applied to a Flow cytometry (Guava easyCyte: Merck). Obtained data were analyzed with an InCyte software (guava soft 3.1.1; Merck). The cell cycle gates were manually adjusted and the percentages of each cell cycle stage (G1-, S-, G2/M-phase or dead cell) were calculated.

**Quantification, statistics, and reproducibility**. The fluorescence signal intensities of GFP-Dsn1, Nuf2-GFP, Nuf2-mScarlet, anti-Bub1, anti-Knl1, VH-mScarlet, and anti-human Ndc80 on more than 60 kinetochores in each of 10 cells (Figs. 1g, 2b, 2d, 2f, 2g-i, 3a, 3b, 4a, 4e, 4h, 5c, 6c and Supplementary Fig. 5b-d) and background signals in nonkinetochore region for each sample were quantified using Imaris v9.0.2. (Bitplane). The mean of fluorescence signal intensities on kinetochores in each cell were subtracted with mean of background signals in non-kinetochore region. *p* values were calculated by two-tailed Welch's *t*-test (Figs. 2d, 2g-I, 4e, 5c and Supplementary Fig. 5b-d) or one-way analysis of variance (ANOVA) followed by Tukey's test (Fig. 1g, 2b, 2f, 3a, 3b, 4a, 4h and 6c).

The fluorescence signal intensities of H3T3ph and Aurora B in the centromere region and the noncentromere region were quantified using Fiji. The centromere region was selected by the 10 × 15 pixels (length × width) region along the chromosome axis (the region of interest: ROI-A). This region was enlarged with the 10 × 20 pixels (length × width) region (ROI-B) for background subtraction. To determine the background signal intensity, the sum of fluorescence signal intensity of ROI-B was subtracted with the sum of fluorescence signal intensity of ROI-A (ROI-B - ROI-A). The mean of fluorescence signal intensity on the centromere region (ROI-A) was subtracted with the mean of the background signal intensity. The noncentromere region was selected by the length size (pixel length size depends on length of chromosome arm) × 15 pixels (length × width) region along with chromosome axis. Background subtraction was performed in the same way as the centromere region. The fluorescence signals on 4 chromosomes in each cell (Fig. 3c: *n* = 14 (WT) and 13 (Δ121–240) cells; Fig. 3d, Supplementary Fig. 5e, 5f: *n* = 13 cells) were quantified. *p* values were calculated by two-tailed Welch's *t*-test.

The kinetochore-kinetochore distances of more than 10 mitotic chromosome in each of 10 cells (Figs. 3e, 5h, Supplementary Fig. 5g) in chromosome spread samples were measured using Imaris v9.0.2. (Bitplane). *p* values were calculated by two-tailed Welch's *t*-test.

The chromosome alignment values were calculated as described in a section for "Evaluation of chromosome alignment in DT40 cells". *p* values were calculated by one-way ANOVA followed by Tukey's test (Figs.1j, 5f, 5l) or two-tailed Welch's *t*-test (Supplementary Fig. 1n) (Fig. 1j: *n* = 41 (None), 46 (WT), and 46 (Δ121-240) cells; 5 f: *n* = 42 (None), 41 (WT), 41 (Δ121-240) and 41 (2X(1-90)_Δ121-240) cells; 5 l: *n* = 37 (Dsn1^WT), 35 (Dsn1^Δ326-349) and 42 (Dsn1^Δ326-349 + CENP-T^2X(1-90)) cells; Supplementary Fig. 1n: *n* = 44 (Nocodazole), and 42 (MG132) cells).

Each experiment was repeated at least two (Figs. 1d, 1e, 1g-j, 2e, 2h, 2i, 3a, 3d, 3e, 4f, 4g, 5c, 5d, 5f-n, 6b-d, 6g and Supplementary Figs. 1c, 1e, 1g, 1i, 1k-o, 2b-e,

2h, 2k-m, 3a, 3b, 4c-e, 5a, 5e-h, 5j, 6a-f, 7c, 7e and 7f) or three times (Figs. 1k, 2b, 2d, 2f, 2g, 3b, 3c, 4a, 4d-e, 4h, 5e and Supplementary Fig. 5b-d). Representative data of technical replicates are presented.

The percentage of cells with misaligned chromosomes (Fig. 6e) or cells with abnormal chromosome (Fig. 6f) were calculated. Each experiment was repeated three times (Fig. 6e: $n = 24$ or 25 cells per experiment) or four times (Fig. 6f: $n = 24$ or 25 cells per experiment), and mean and standard deviation of three (Fig. 6e) or four (Fig. 6f) experiments are presented. $p$ values were calculated by one-way ANOVA followed by Tukey's test.

Statistical analyses were performed using GraphPad Prism7(GraphPad Software).

**Reporting summary**. Further information on research design is available in the Nature Research Reporting Summary linked to this article.

## Data availability
Source data are provided with this paper. All data supporting the findings of this study are available within the paper and supplementary information. Source data are provided with this paper.

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

## Acknowledgements
The authors are very grateful to Yuko Fukagawa, Kaori Oshimo, Yumiko Takeshita and Reika Fukuoka for their technical assistance. We also thank Thomas Maresca for providing plasmid constructs. This work was supported by CREST of JST (JPMJCR21E6), JSPS KAKENHI Grant Numbers 16H06279, and 17H06167, 20H05389, 21H05752 to TF, JSPS KAKENHI Grant Numbers 16K18491, and 21H02461 to MH.

## Author contributions
Y.T. designed and performed all experiments in this study under the supervision of M.H. and T.F. T.F. wrote the manuscript in collaboration with Y.T., discussing it with M.H.

## Competing interests
The authors declare no competing interests.
