## [Peer Review File · Nature Communications]

Recruitment of two Ndc80 complexes via the CENP-T pathway is sufficient for kinetochore functionsREVIEWER COMMENTS

Reviewer #1 (Remarks to the Author):

Proper segregation of chromosomes crucially depends on the multisubunit KMN complex; in many cell types it becomes recruited to kinetochore via two scaffolds: CENP-T and CENP-C. Whether different modes of KMN recruitment, and specifically of its microtubule-binding component Ndc80, have different biological roles is an unresolved outstanding question in the field. This paper builds on a significant body of work and tools generated previously in the Fukagawa lab to study the assembly and regulation of vertebrate kinetochore. The authors take advantage of chicken cells, where deletion of the N-tail of CENP-C was previously found to be non-essential for growth and propagation. The CENP-T-KMN interaction, however, is essential. In the absence of CENP-C tail, Ndc80 binding through the CENP-T pathway is sufficient for chromosome segregation, allowing the authors to dissect this pathway in details. In the current paper, the authors devise a set of clever molecular tools to specifically perturb Ndc80 binding sites at CENP-T: its direct binding site and through the Mis12 complex.

The authors conclude that “the Ndc80 complex and Mis12 complex are separable in the KMN network on the CENP-T pathway”. In my opinion the word “separable” is not a good choice to describe the main results, and this title might be confusing to non-specialists. It has long been known that Knl1 branch of the KMN complex is engaged mostly in the checkpoint and other regulatory functions, whereas the Ndc80 is responsible for microtubule binding, so in this sense these components are functionally separable. In my opinion, the main novel finding in this paper is that in DT40 cells disrupting the Ndc80 binding to Mis12 site in the KMN is lethal but it can be rescued by engineering the second direct binding site for Ndc80 at the CENP-T tail. The authors are encouraged to seek a different title that reflects their results more accurately or define better what they mean by “separable”. Although this MS has some interesting new findings, many conclusions are in my opinion premature and the MS is not suitable for publication in NCB in the present form.

Major concerns:

1) Insufficiently rigorous evaluation of the Ndc80 role in kinetochore-microtubule interactions. The main read-out for Ndc80 function in the manuscript is cell viability (Figures 4f; 5f-g; 6c). Viability is not a very informative criterion for pathways dissection, and such crude method is not acceptable for deducing fine structure-function relationships. More informative tests should be used to evaluate kinetochore functions of Ndc80 in microtubule binding and error correction (STLC washout, inter-kinetochore distance, lagging chromosomes and chromosome loss in anaphase would be examples of commonly used functional assays). It is possible that different genetic perturbations that leave cells viable, may have different impact on accuracy of chromosome segregation. Other specific suggestions are provided below.

a. The authors state repeatedly that two copies of Ndc80 are needed for proper kinetochore-microtubule interactions and maintenance of spindle integrity. However, many cell images appear to contradict this statement. Figures 1f; 2b,f shows cell lines with no direct Ndc80 binding sites (CENP- Δ 73 and depleted CENP-T, and CENP- Δ 73 with CENP-T Δ 90), which should result in zero Ndc80 complexes at kinetochore. Contrary to the authors claim, the chromosomes in these cells appear to align in metaphase plate. More convincing evidence should be provided, both images and quantifications to evaluate phenotypes of examines cell lines. Live cell imaging experiments would be ideal for understanding the phenotype in cells that differ in mode of Ndc80 tethering.

b. On page 13, the authors concluded that: "As expected, we observed proper kinetochore microtubule attachments and normal bipolar spindle shapes in cKO-CTCC Δ 73/CENP-T2x(1-90) Δ 121-240 cells (Figure 5d), suggesting that the Ndc80C function in cKO-CT-CC Δ 73/CENP-T Δ 121-240 cells was rescued by the addition of extra Ndc80C-binding sites." This is a very strong statement which is central to this work, but it is not backed up by convincing quantifications. Authors should examine Ndc80 function using established functional assays, e.g. inter kinetochore distance, percent of cells with bipolar mitotic spindle, fraction of cells in mitosis, etc.

c. On page 14, the authors concluded that KM and N-N (two copies of Ndc80C) are separable and have distinct essential functions for cell growth based on the comparison of viability of cell lines GFP-CENP-T 2x(1-90) cKO-Dsn1/Dsn1 Δ 326-349 cell vs. CENP-T2x(1-90) Δ 121-240 cells (Figure 5g). To strengthen this conclusion, the authors should demonstrate directly that cells with GFP-CENP-T 2x(1-90) cKO-Dsn1/Dsn1 Δ 326-349 do not have defects in microtubule binding, cohesion and spindle assembly, and add these data to Figures 5d,e.

d. On page 12, the authors evaluate Ndc80 function indirectly by measuring tension exerted through CENP-T. They conclude that microtubules fail to apply tension to CENP-T in the cKO-Dsn1 Δ 326-349 (Tet-Off)/CENP-T-TR/VH-mScarlet cells (Figure 4h). However, the presented data are normalized, which precludes the direct comparison between WT and Δ 326-349 cell lines. Full information should be provided.

e. Experiments with RPE-1 cells are very important since they aim to establish whether conclusions drawn from DT40 are applicable to other cell types. On page 14 the authors conclude that "the Mis12C-Ndc80C interaction is dispensable in human cells". The authors base it only on measurements of Ndc80 levels and viability in RPE-1 cells mCherry-DSN1 WT or mCherry-DSN1 Δ 325-356. I think that the key unanswered question is whether the mutant cells have any mitotic defects. To strengthen the MS, the authors should carry out analysis of kinetochore-microtubules interactions and error correction in these cells. Also, they should provide more direct evidence for the lack of binding between kinetochore complexes in these cells, e.g. by immunoprecipitation of Mis12 complex from RPE-1 cells cKO-hDsn1 / Dsn1 Δ 325-356 and cKO-hDsn1 / Dsn1 WT) followed by western blotting.

f. The authors show that removing the Ndc80 binding site on Mis12 in human cells is not lethal, which is in contrast to the results of analogous perturbations in DT40 cells. The authors interpret this result by suggesting that chromosome segregation in all organisms requires two Ndc80 copies per each scaffolding protein at the kinetochore. I am not convinced by this statement as other interpretations are possible. Obviously, too little Ndc80 molecules per kinetochore may not be sufficient for its segregation, but there is no evidence that exactly two molecules are required (as oppose to some other number like 1.5 . Also the authors only estimate average number of molecules, and some binding sites may be unoccupied). I think that statements about the requirement of two Ndc80 molecules should be amended to reflect the lack of information about the exact roles of differently tethered Ndc80 complexes at the kinetochore, but also because current quantifications of Ndc80 occupancy have many problems (see below).

g. The authors state repeatedly that “single copy of Ndc80C to spindle microtubules is not enough to generate a force to pull a chromosome”, meaning single Ndc80 attached to CENP-T. This statement is found in the current MS and prior publications by this group, but it is not accurate. First, Ndc80 does not generate force since it is not a molecular motor. Second, it is presently unknown how different Ndc80 molecules transduce force from microtubule depolymerization. Current in vitro assays measure ability of these molecules to couple to dynamics microtubule ends or to withstand rupture force; these are very different force-dependent features. Phenotypically, they could be assessed in cells by observing chromosome segregation in anaphase, but making direct conclusions about force magnitude is hard, if not impossible with current tools. The authors should provide more functional data to support their conclusion or to rewrite the text to align it better with the current state of knowledge.

2) Quantifications of Ndc80, Mis12 and Knl1 levels at kinetochores raise concerns.

a. Estimates of the Ndc80 level at kinetochore rely strongly on the authors' ability to remove endogenous protein, such as WT CENP-T. However, Figures 1f; 2b; 2f; 4a show that kinetochores in cells with CENP- Δ 73 and depleted CENP-T still have some associated Ndc80 and Mis12, suggesting a significant presence of residual CENP-T protein. Immunoblot in Figure S5a also does not show complete depletion of Tet-Dsn1 in the cell line with no expression of Dsn1 transgene (“None”) in the presence of tetracycline. Authors should repeat this and other key experiments under conditions when endogenous protein is completely deleted (immunoblots should be provided).

b. The above issue is a significant problem for estimating levels of kinetochore proteins because current quantifications do not take into account the baseline level of Ndc80 and Mis12 on the kinetochore without expression of CENP-T transgene (“None”) (Figures 1f; 2b,f; 3a-b; 4a). Consequently, authors' conclusions about Ndc80 level are not convincing. (For example, in the following sentences: “As shown in Figure 4a, Ndc80C levels in these cells were reduced to half of those in cKO-CT-CC Δ 73/CENP-TWT

cells, indicating that one of the roles of KM is the recruitment of Ndc80C... Our model suggests that one copy of Ndc80C still binds to the N-terminal region of CENP-T Δ 121-240.”).

c. Experiment described in Figure 4e led the authors to conclude that “Ndc80C (Nuf2-mScarlet) levels in cKO-Dsn1/Dsn1 Δ 326-349 cells were reduced to approximately half that of the control cells” (page 12). However, this experiment lacks data for cells with no expression of CENP-T transgene (“None”). The baseline level of Ndc80 at kinetochore should be shown because it may change this conclusion.

d. Figure 3a shows that KNL1 level is higher when there is no expression of CENP-T (“None”) than in the cells with deleted Mis12 binding site (“ Δ 121-240”). This is unexpected because both cell lines should have no Knl1 at kinetochores. This issue should be addressed.

e. Figure legends lack sufficient description of how data are presented. In most cases it is unclear whether single data point represents results from one cell, one kinetochore, or average of many measurements. Legends should be improved and full statistics should be disclosed in figure legends and source data file.

3) Effect of tension on recruitment of Ndc80 and Mis12 to CENP-T is not well documented. The authors state on page 9 that the tension exerted by kinetochore microtubules has no effect on Mis12C recruitment onto CENP-T. This conclusion is not convincing because the authors do not provide any measurements to evaluate kinetochore tension. Moreover, cell images in Figure 2 h and i do not readily reveal significant differences in cells treated with different drugs, as chromosomes appear unaligned after MG132 treatment. I am concerned that kinetochore tension was not generated in these cells, explain why they were not significantly different from nocodazole-treated cells. Please provide direct evidence for interkinetochore tension (e.g. inter kinetochore distance) in these and control cells.

4) Chromosome cohesion phenotype. Authors use interkinetochore distance in isolated chromosomes to support their claim that significant cohesion defects are present in cells cKO-CT-CC Δ 73 (Tet-Off) expressing CENP-T Δ 121-240 or lacking Mis12 binding site or CENP-T 2x(1-90) Δ 121-240 with artificial Ndc80 binding site (Figure 3e; 5e). Can this difference be caused by defects in chromatin condensation or some other issues? More direct assays should be used to demonstrate reduced cohesion and/or reduced level of cohesins.

Minor comments:

1) Figures

a. Figures 1b; 2b,d,f; 3a,b; 4a; 5c. mScarletCENP-C Δ 73 is localized not only on kinetochores, but it also distributed along chromosome arms. Has this been seen before in DT40 or other cells? The authors should provide an explanation or discuss these results.

b. Figure 1g. Please show more time points to demonstrate, that cKO-CT-CC Δ 73 (Tet-Off) /CENP-T Δ 161-200 and cKO-CT-CC Δ 73 (Tet-Off) /CENP-T Δ 201-216 are dead. The difference with one time point is not enough for this important conclusion.

c. Figure 3c. Please provide more context for the observed increase of H3T3ph in cells with no Mis12. A brief discussion of this effect and references to prior publications would help to understand the importance of this observation.

d. Figure 3d. Selected images of cells do not correspond to quantifications. In WT cell image the level of Aurora B at the centromere is significantly higher than seen for CENP-T Δ 121-240 cell, whereas the quantification shows no statistical difference between these cells. The authors should provide more representative images.

e. Figure S4b,c. Both panels show results for cKO-Nsl1 (AID) cells. In panel b level of GFP-AID-Nsl1 is auxin-dependent, however, analogous results in panel c show a lack of auxin dependency. Authors should explain the design of this experiment (include diagram as in Figure S1) and why different results are seen in these two panels.

2) The authors could be more scholarly in their citation of the literature.

a) Some important statements in introduction are not supported by references. For example: "Furthermore, in addition to Knl1C-Mis12C-Ndc80C (KMN) formation, an additional copy of Ndc80C directly binds to CENP-T, and one KMN unit and one Ndc80C forms KMN-N in the network on CENP-T.

The KMN network appears to play multiple roles, including in the binding of microtubules, spindle checkpoint functions, and cohesion of sister chromatids.”

b) Current paper shows that Ndc80 binding to CENP-T is required for Mis12 recruitment in cells. This has been shown previously (Hori et al., 2013; Rago et al, 2015). Proper discussion and references should be included to clarify which aspect of this result is considered novel by the authors, or this section should be toned down.

c) Page 5 and in discussion: “.. the Ndc80C-Mis12C interaction is thought to be critical for the kinetochore functions..” Please improve citations. Alushin et al 2010 and DeLuca et al. 2006 do not have relevant data.

3) Legends to some figures do not explain which data are presented with “none” label. Please improve.

4) Page 12, line 9; page 13, lines 7,16; page 31 line 1. Authors claim that they perform “k-fiber microtubule staining”. This is misleading, they perform staining of all spindle microtubules.

Reviewer #2 (Remarks to the Author):

Faithful chromosome segregation is achieved thanks to the KMN network seeded on centromeres by two major pathways depending on either CENP-C or CENP-T. Their relative contributions to kinetochore function differ among species, and their redundancy is debated in the field. Here, Takenoshita et al tested the functional importance of distinct KMN sub-complexes recruited by CENP-T. Using the chicken DT40 cell system, they designed a comprehensive set of cell lines with various mutant CENP-C and CENP-T alleles expressed under the endogenous promoters or under a Tet-Off promoter, which allowed them to dissect the contribution of each pathway to KMN network formation. They demonstrated that Knl1C and Mis12C (KM) components of the KMN network are functionally separate from Ndc80C (N). KM is necessary for localizing key mitotic kinases to centromeres and for centromere cohesion. Severing the KM-N interaction reduced Ndc80C levels at kinetochores and caused cell death, which was rescued by tethering additional copies of Ndc80C directly to CENP-T. The authors concluded that the Mis12C-Ndc80C interaction is not essential, and that KM and N components of KMN network are functionally distinct. Although the conclusions are largely consistent with previous work, the paper represents a significant addition to the kinetochore field as a thorough, precise, detailed analysis of the CENP-T pathway, taking advantage of state-of-the art genetic complementation in the DT40 cell system. The data are high quality, and the results support the conclusions without requiring additional experiments, but the paper could benefit from some textual editing to improve clarity.

Comments:

1. The number of repetitions performed for each experiment should be indicated throughout the paper. Fig 1e, for example, is presented as motivation for the later analyses, so it is important to know how much variation is observed in between repetitions.
2. Fig. 1d - The band marked by an asterisk could potentially be contamination with cells expressing 3xFLAG CENP-T WT, as the band migrates at exactly the same size. Can the authors use a different FLAG antibody exclude this possibility?
3. Fig. S1a – The authors should explain the reason for choosing the b-actin locus for expression of mScarlet-CENP-C .
4. Fig. 1d and Fig. S2c – It looks like mScarlet-CENP-C migrates differently in cells without CENP-T expression or in cells expressing CENP-Tdelta90. Can the authors comment on this?
5. Fig 3e – It's difficult to see any difference in inter-kinetochore distance between the two images shown. It would help to measure the distances in these two examples and include that measurement in the figure to show that the examples are representative of the quantitation in the bar graph.
6. Results (p 11) – “However, as in our previous model 37, a half reduction of Ndc80C levels on CENP-T was observed.” This sentence seems redundant with the previous sentence. If it is making an additional point, that should be explained more clearly.
7. Results (p 14) – “Strikingly, expression of GFP-CENP-T2x(1-90) in cKO-Dsn1/Dsn1Δ326-349 cells completely compensated for growth defects in the cKO-Dsn1/Dsn1Δ326-349 cells in the presence of Tet (Figure 5g and S6d). These data indicate that Ndc80C does not necessarily bind to Mis12C.” If I understand the authors' point here, the second sentence should be worded differently, such as “... does not necessarily need to bind ...” or the conclusion should be explained more clearly.

8. Fig. 6a – The CENP-C pathway should be included in the schematics because Dsn1 mutation affects CENP-C-KMN as well as CENP-T-KMN. Therefore, the phenotypes measured in this experiment will depend on changes in Ndc80 recruitment by both CENP-C and CENP-T pathways.

9. Discussion (p 16) – “Two copies of Ndc80C (N-N) are necessary and sufficient for proper kinetochore-microtubule attachment in DT40 cells, suggesting that binding of a single copy of Ndc80C to spindle microtubules is not enough to generate a force to pull a chromosome. Single-molecule experiments demonstrated that the multivalency of Ndc80C efficiently tracked depolymerizing microtubules, whereas a single copy of Ndc80C did not, which supports our observations.” This statement is misleading. Even with a single copy of Ndc80 per CENP-T, multivalent interactions could still occur due to multiple copies of CENP-T (and therefore multiple Ndc80 molecules) at each kinetochore. Can the authors speculate about why multiple copies of Ndc80 may need to bind to a single CENP-T?

Reviewer #3 (Remarks to the Author):

The kinetochore is a complex organelle that is assembled in distinct protein layers to move chromosomes and to establish a platform for SAC signaling. The functional interface of the kinetochore consists of two critical protein components: the Ndc80 complex and the Knl1-Mis12 (KM) protein network. In some eukaryotes, this interface is assembled by two centromeric proteins: CENP-T and CENP-C. Using a series of elegant, cell biological experiments, Takenoshita and colleagues clarify the roles of Ndc80 and KM molecules that are recruited by CENP-C and CENP-T play in kinetochores in the DT40 chicken cells. They provide several insights into the assembly mechanisms of the DT40 kinetochores. (1) The recruitment of at least two Ndc80 molecules per CENP-T is essential for the formation of proper kinetochore-microtubule attachment. (2) The two Ndc80 molecules per CENP-T can either be recruited via direct interaction with CENP-T or one via direct interaction and the other via Mis12 complex bound to CENP-T. (3) The recruitment and functions of Ndc80 and KM molecules recruited by CENP-T are separable in DT40 kinetochores.

These key findings clarify the inter-dependencies between the recruitment of Ndc80 complex and KM network molecules and show that their functions are separable. This is a key advance in our understanding of the eukaryotic kinetochore. The analysis presented by the authors is rigorous and extensive. For these reasons, I wholeheartedly support the publication of this manuscript with only minor changes to the manuscript. I do not think that any additional experiments are necessary.

1. In their cell growth assays, the authors describe reduced growth as a “growth delay” (e.g., at the top of page 7). This description isn’t quite accurate, because the retarded growth most likely results from increased cell death.
2. Page 8, 2nd para: “We examined other phosphorylation....” – these sentences need to be edited for clarity – e.g., “We examined the effects of CENP-T phosphorylation on Mis12 recruitment. There are eight phosphorylatable residues...”
3. Page 10, end of 2nd para: “we found that the distances between sister kinetochore pairs were increased” should be changed to: “distance between sister kinetochores increased”
4. Page 10, last para: “cohesion of sister chromatids or the recruitment of checkpoint proteins” should be changed to: cohesion of sister chromatids and the recruitment of checkpoint proteins
5. Page 11, 2nd para: “Although we found that one copy of Ndc80C still exists” should be changed to: “Although we found that one copy of Ndc80C per CENP-T still exists”
6. It will be useful to add key details and the citation for the tension sensor probe in the main text, because there is also a FRET-based probe used to assess tension
7. Page 14, top: “These data indicate that Ndc80C does not necessarily bind to Mis12C” This is a very striking finding and needs to be explained more clearly. E.g., “Ndc80 need not interact with Mis12 for establishing MT-binding”. It might be a good idea to cite in vitro work from the Davis lab on yeast Ndc80-Mtw1 complex in the discussion, which suggests regulation of the MT-binding activity of Ndc80 through the Ndc80-Mtw1 interaction (PNAS).

Ajit Joglekar

Response to Reviewers Takenoshita et al. (NCOMMS-21-26789-T)

Reviewer #1

Proper segregation of chromosomes crucially depends on the multisubunit KMN complex; in many cell types it becomes recruited to kinetochore via two scaffolds: CENP-T and CENP-C. Whether different modes of KMN recruitment, and specifically of its microtubule-binding component Ndc80, have different biological roles is an unresolved outstanding question in the field. This paper builds on a significant body of work and tools generated previously in the Fukagawa lab to study the assembly and regulation of vertebrate kinetochore. The authors take advantage of chicken cells, where deletion of the N-tail of CENP-C was previously found to be non-essential for growth and propagation. The CENP-T-KMN interaction, however, is essential. In the absence of CENP-C tail, Ndc80 binding through the CENP-T pathway is sufficient for chromosome segregation, allowing the authors to dissect this pathway in details. In the current paper, the authors devise a set of clever molecular tools to specifically perturb Ndc80 binding sites at CENP-T: its direct binding site and through the Mis12 complex.

The authors conclude that “the Ndc80 complex and Mis12 complex are separable in the KMN network on the CENP-T pathway”. In my opinion the word “separable” is not a good choice to describe the main results, and this title might be confusing to non-specialists. It has long been known that Knl1 branch of the KMN complex is engaged mostly in the checkpoint and other regulatory functions, whereas the Ndc80 is responsible for microtubule binding, so in this sense these components are functionally separable. In my opinion, the main novel finding in this paper is that in DT40 cells disrupting the Ndc80 binding to Mis12 site in the KMN is lethal but it can be rescued by engineering the second direct binding site for Ndc80 at the CENP-T tail. The authors are encouraged to seek a different title that reflects their results more accurately or define better what they mean by “separable”. Although this MS has some interesting new findings, many conclusions are in my opinion premature and the MS is not suitable for publication in NCB in the present form.

We thank you for the appreciation of our work. According to your suggestions, we have changed the title to “The dual Ndc80 complex on CENP-T functions without direct binding to the Mis12 complex” We believe that the new title reflects our main conclusions. In addition to changing the title, we tried to do our best to address all concerns raised by you.

Major concerns:

- 1) Insufficiently rigorous evaluation of the Ndc80 role in kinetochore-microtubule interactions. The main read-out for Ndc80 function in the manuscript is cell viability (Figures 4f; 5f-g; 6c). Viability is not a very informative criterion for pathways dissection, and such crude method is not acceptable for deducing fine structure-function relationships. More informative tests should be used to evaluate kinetochore functions of Ndc80 in microtubule binding and error correction (STLC washout, inter-kinetochore distance, lagging chromosomes and chromosome loss in anaphase would be examples of commonly used functional assays). It is possible that different genetic perturbations that leave cells viable, may have different impact on accuracy of chromosome segregation. Other specific suggestions are provided below.*

We agree with this comment. As you suggested, the cell viability test was not sufficient to assess the function of the kinetochore. We have added data for cell-cycle distribution, rate of chromosome alignment, and rate of the bipolar spindle for each analysis in the revised version (Figures 1, 5, and 6).

- a. The authors state repeatedly that two copies of Ndc80 are needed for proper kinetochore-microtubule interactions and maintenance of spindle integrity. However, many cell images appear to contradict this statement. Figures 1f; 2b,f shows cell lines with no direct Ndc80 binding sites (CENP-C Δ 73 and depleted*

CENP-T, and CENP-CΔ73 with CENP-TΔ90), which should result in zero Ndc80 complexes at kinetochore. Contrary to the authors claim, the chromosomes in these cells appear to align in metaphase plate. More convincing evidence should be provided, both images and quantifications to evaluate phenotypes of examines cell lines. Live cell imaging experiments would be ideal for understanding the phenotype in cells that differ in mode of Ndc80 tethering.

We agree that our data presentation was ambiguous. In some Figures, images presenting aligned chromosomes in mutant cells were used. Many chromosomes were not aligned at the metaphase plate in a large population of mutant cells lacking Ndc80 binding sites. In the revised version, we have demonstrated that the chromosome alignment rate and observed that chromosomes in cKO-CT-CCΔ73/CENP-TΔ121-240 (Figure 1j) or cKO-Dsn1/Dsn1Δ326-349 (Figure 5k) cells were not well-aligned. We have also presented misaligned chromosome images in the revised version. However, we previously presented that the amount of Mis12C at kinetochores changes during mitosis (Hara et al., Nature Cell Biol., 2018). Therefore, we must measure Mis12C levels at aligned chromosomes for accurate quantification at the same cell cycle stages.

b. On page 13, the authors concluded that: “As expected, we observed proper kinetochore microtubule attachments and normal bipolar spindle shapes in cKO-CTCCΔ73/CENP-T2x(1-90)_Δ121-240 cells (Figure 5d), suggesting that the Ndc80C function in cKO-CT-CCΔ73/CENP-TΔ121-240 cells was rescued by the addition of extra Ndc80C-binding sites.” This is a very strong statement which is central to this work, but it is not backed up by convincing quantifications. Authors should examine Ndc80 function using established functional assays, e.g. inter kinetochore distance, percent of cells with bipolar mitotic spindle, fraction of cells in mitosis, etc.

We appreciate and agree with this comment. Since the experiments you suggested are very important, we measured the rate of cells with the bipolar mitotic spindle and cell-cycle distribution in cKO-CT-CCΔ73/CENP-T 2x(1-90)_Δ121-240 cells. We have added a new data to Figure 5 (new Figure 5e, f, and g) in the revised version.

c. On page 14, the authors concluded that KM and N-N (two copies of Ndc80C) are separable and have distinct essential functions for cell growth based on the comparison of viability of cell lines GFP-CENP-T 2x(1-90) cKO-Dsn1/Dsn1Δ326-349 cell vs. CENP-T2x(1-90)_Δ121-240 cells (Figure 5g). To strengthen this conclusion, the authors should demonstrate directly that cells with GFP-CENP-T 2x(1-90) cKO-Dsn1/Dsn1Δ326-349 do not have defects in microtubule binding, cohesion and spindle assembly, and add these data to Figures 5d,e.

We agree with this comment and performed several experiments to evaluate the mitotic functions of GFP-CENP-T 2x(1-90) cKO-Dsn1/Dsn1Δ326-349 cells. We have added the data for the spindle shape, numbers of bipolar spindles, chromosome alignments, and cell cycle distribution to Figure 5 (new Figure 5i, j, k, l) in the revised manuscript. These data strongly support our conclusion that expression of CENP-T 2x(1-90) suppresses the deficiency in cKO-Dsn1/Dsn1Δ326-349 cells.

d. On page 12, the authors evaluate Ndc80 function indirectly by measuring tension exerted through CENP-T. They conclude that microtubules fail to apply tension to CENP-T in the cKO-Dsn1 Δ326-349 (Tet-Off)/CENP-T-TR/VH-mScarlet cells (Figure 4h). However, the presented data are normalized, which precludes the direct comparison between WT and Δ326-349 cell lines. Full information should be provided.

We agree with this comment. We have now presented full information for tension-sensor assay in the revised Figure 4h.

e. Experiments with RPE-1 cells are very important since they aim to establish whether conclusions drawn from DT40 are applicable to other cell types. On page 14 the authors conclude that “the Mis12C-Ndc80C interaction is dispensable in human cells”. The authors base it only on measurements of Ndc80 levels and viability in RPE-1 cells mCherry-DSN1 WT or mCherry-DSN1 Δ 325-356. I think that the key unanswered question is whether the mutant cells have any mitotic defects. To strengthen the MS, the authors should carry out analysis of kinetochore-microtubules interactions and error correction in these cells. Also, they should provide more direct evidence for the lack of binding between kinetochore complexes in these cells, e.g. by immunoprecipitation of Mis12 complex from RPE-1 cells cKO-hDsn1 / Dsn1 Δ 325-356 and cKO-hDsn1 / Dsn1 WT) followed by western blotting.

Like analyses for other DT40 mutant cells, we have added new data to evaluate mitotic function in human RPE-1 cells expressing mScarlet-Dsn1 WT or mScarlet-Dsn1 Δ 325-356. We demonstrated the spindle shape, chromosome alignment, and rate of lagging chromosomes in these cells (new Figure 6d, e, f). In addition, according to your suggestions, we performed immunoprecipitation with anti-RFP and demonstrated that Ndc80 did not co-precipitate with the mutant Dsn1 in cells expressing mScarlet-Dsn1 Δ 325-356 by IP-immunoblot analysis (new Figure 6b).

f. The authors show that removing the Ndc80 binding site on Mis12 in human cells is not lethal, which is in contrast to the results of analogous perturbations in DT40 cells. The authors interpret this result by suggesting that chromosome segregation in all organisms requires two Ndc80 copies per each scaffolding protein at the kinetochore. I am not convinced by this statement as other interpretations are possible. Obviously, too little Ndc80 molecules per kinetochore may not be sufficient for its segregation, but there is no evidence that exactly two molecules are required (as oppose to some other number like 1.5 . Also the authors only estimate average number of molecules, and some binding sites may be unoccupied). I think that statements about the requirement of two Ndc80 molecules should be amended to reflect the lack of information about the exact roles of differently tethered Ndc80 complexes at the kinetochore, but also because current quantifications of Ndc80 occupancy have many problems (see below).

We agree with your comment, and we did not conclude that two Ndc80 copies are required for chromosome segregation. While we preferentially propose our model, we have described other possibilities to explain this phenotype in the revised text.

g. The authors state repeatedly that “single copy of Ndc80C to spindle microtubules is not enough to generate a force to pull a chromosome”, meaning single Ndc80 attached to CENP-T. This statement is found in the current MS and prior publications by this group, but it is not accurate. First, Ndc80 does not generate force since it is not a molecular motor. Second, it is presently unknown how different Ndc80 molecules transduce force from microtubule depolymerization. Current in vitro assays measure ability of these molecules to couple to dynamics microtubule ends or to withstand rupture force; these are very different force-dependent features. Phenotypically, they could be assessed in cells by observing chromosome segregation in anaphase, but making direct conclusions about force magnitude is hard, if not impossible with current tools. The authors should provide more functional data to support their conclusion or to rewrite the text to align it better with the current state of knowledge.

We agree with your comment that Ndc80 does not generate a force. We have carefully revised these statements and have not mentioned “Ndc80C generates force” in the revised text.

2) Quantifications of Ndc80, Mis12 and Knl1 levels at kinetochores raise concerns.

a. Estimates of the Ndc80 level at kinetochore rely strongly on the authors’ ability to remove endogenous protein, such as WT CENP-T. However, Figures 1f; 2b; 2f; 4a

show that kinetochores in cells with CENP-C Δ 73 and depleted CENP-T still have some associated Ndc80 and Mis12, suggesting a significant presence of residual CENP-T protein. Immunoblot in Figure S5a also does not show complete depletion of Tet-Dsn1 in the cell line with no expression of Dsn1 transgene ("None") in the presence of tetracycline. Authors should repeat this and other key experiments under conditions when endogenous protein is completely deleted (immunoblots should be provided).

We appreciate this comment. In CENP-T (or Dsn1) conditional knockout cells, cells died after being arrested at mitosis upon Tet addition before Tet-response proteins (CENP-T or Dsn1) were completely lost, although protein amounts were dramatically reduced. We repetitively performed immunoblot experiments (see below for cKO Dsn1 cells) and have confirmed these results. We have emphasized that these Tet-response proteins were completely lost when we rescued cell death by expression of the exogenous wild-type protein (lane 5, WT), suggesting that the Tet system was working well. Therefore, Tet-responsive proteins could not be completely depleted in CENP-T or Dsn1 conditional knockout cells. By immunofluorescence analysis, CENP-T proteins were not detected in cKO-CENP-T cells, albeit were visible when we enhanced the contrast (see images below). Therefore, we needed to measure the Ndc80C levels for CENP-T or Dsn1 conditional knockout cells as baseline control ("None"). In the revised version, we have presented raw data and normalized

data using data from baseline controls (Figure 1g, 2b, 2f, 3a, 3b, 4a). We have also explained that we detected residual proteins in conditional knockout cells as a feature of these cells, in the revised version.

b. The above issue is a significant problem for estimating levels of kinetochore proteins because current quantifications do not take into account the baseline level of Ndc80 and Mis12 on the kinetochore without expression of CENP-T transgene ("None") (Figures 1f; 2b-f; 3a-b; 4a). Consequently, authors' conclusions about Ndc80 level are not convincing. (For example, in the following sentences: "As shown in Figure 4a, Ndc80C levels in these cells were reduced to half of those in cKO-CT-CC Δ 73/CENP-TWT cells, indicating that one of the roles of KM is the recruitment of Ndc80C... Our model suggests that one copy of Ndc80C still binds to the N-terminal region of CENP-T Δ 121-240.")

As we explained in the comment (2-a), we presented both raw and normalized data using data of cells without expression of the transgene ("None") as baseline control (new figure 1g, 2b, 2f, 3a, 3b, 4a). Since conditional knockout cells were arrested at mitosis before cell death, Tet-responsive proteins were not completely lost. We have further used data from these cells as a baseline level. We have also carefully changed the description of the Ndc80C levels in the revised version.

c. Experiment described in Figure 4e led the authors to conclude that "Ndc80C (Nuf2-mScarlet) levels in cKO-Dsn1/Dsn1 Δ 326-349 cells were reduced to approximately half that of the control cells" (page 12). However, this experiment lacks data for cells with no expression of CENP-T transgene ("None"). The baseline level of Ndc80 at kinetochore should be shown because it may change this conclusion.

As you suggested, we could not mention that Ndc80C levels in cKO-Dsn1/Dsn1 Δ 326-349 cells were reduced to approximately half that of the

control cells. We have modified this sentence in the revised manuscript. We would like to emphasize that Ndc80C is still localized to kinetochores in both cKO-Dsn1/Dsn1 Δ 326-349 and cKO-Dsn1 cells since Ndc80C directly binds to CENP-T. In addition, Ndc80C levels in cKO-CENP-T cells and cKO-Dsn1 cells were different, and it was difficult to determine the baseline levels of Ndc80C. We did not mention “half” without a clear baseline.

d. Figure 3a shows that KNL1 level is higher when there is no expression of CENP-T ("None") than in the cells with deleted Mis12 binding site (" Δ 121-240"). This is unexpected because both cell lines should have no Knl1 at kinetochores. This issue should be addressed.

These experiments were performed several times. KNL1 levels in CENP-T knockout cells or cells expressing CENP-T Δ 121-240 were similar, suggesting that there was almost no KNL1 in these cells. However, owing to handling errors, the average values of KNL1 signals in CENP-T knockout cells were slightly higher than those in cells CENP-T Δ 121-240. We have provided new data for Figure 3a in the revised version.

e. Figure legends lack sufficient description of how data are presented. In most cases it is unclear whether single data point represents results from one cell, one kinetochore, or average of many measurements. Legends should be improved and full statistics should be disclosed in figure legends and source data file.

As per your suggestion, we have added more detailed information to the revised Figure legends.

3) Effect of tension on recruitment of Ndc80 and Mis12 to CENP-T is not well documented. The authors state on page 9 that the tension exerted by kinetochore microtubules has no effect on Mis12C recruitment onto CENP-T. This conclusion is not convincing because the authors do not provide any measurements to evaluate kinetochore tension. Moreover, cell images in Figure 2 h and i do not readily reveal significant differences in cells treated with different drugs, as chromosomes appear unaligned after MG132 treatment. I am concerned that kinetochore tension was not generated in these cells, explain why they were not significantly different from nocodazole-treated cells. Please provide direct evidence for interkinetochore tension (e.g. inter kinetochore distance) in these and control cells.

For this assay, we treated cells with nocodazole or MG132, and further added hypotonic buffer to the cells. Therefore, chromosomes did not seem to be aligned in an image of MG132-treated cells. However, we have confirmed that chromosomes were aligned prior to the addition of hypotonic buffer in MG132-treated cells (revised Figure S2). We have added new data and a detailed explanation of this experiment in the revised version.

4) Chromosome cohesion phenotype. Authors use interkinetochore distance in isolated chromosomes to support their claim that significant cohesion defects are present in cells cKO-CT-CC Δ 173 (Tet-Off) expressing CENP-T Δ 121-240 or lacking Mis12 binding site or CENP-T 2x(1-90) Δ 121-240 with artificial Ndc80 binding site (Figure 3e; 5e). Can this difference be caused by defects in chromatin condensation or some other issues? More direct assays should be used to demonstrate reduced cohesion an/or reduced level of cohesins.

We interpreted the increase in the inter-kinetochore distance as a cohesion defect. However, we could not rule out other possibilities. As you suggested, an increase in the inter-kinetochore distance might have been caused by other factors such as defects in chromatin condensation. To avoid this, we tried to select chromosomes of similar sizes to measure the sister-kinetochore distance. In addition, we did not observe an increase in the sister-kinetochore distance in cKO-Dsn1/Dsn1 Δ 326-349 cells (Figure S5g). Although highly condensed mitotic

chromosomes were accumulated in cKO-Dsn1/Dsn1 Δ 326-349 cells, the sister-kinetochore distance did not change, which indicates that condensed chromosomes do not simply increase the sister-kinetochore distance. We also measured the intensities of Sgo1 at centromeres using Sgo1-GFP knock-in cell lines. However, as presented in the attached images (below), the signals vary in each chromosome. Since it was difficult to evaluate signal intensities for Sgo1, we could not compare Sgo1-level between cells expressing CENP-T WT and CENP-T Δ 121-240.

Minor comments:

1) *Figures*

a. Figures 1b; 2b,d,f; 3a,b; 4a; 5c. mScarletCENP-C Δ 73 is localized not only on kinetochores, but it also distributed along chromosome arms. Has this been seen before in DT40 or other cells? The authors should provide an explanation or discuss these results.

We introduced an mScarletCENP-C Δ 73 gene into the β -actin locus, and its expression levels were higher than those of endogenous CENP-C. These high expression levels caused mislocalization in some CENP-C populations. We have described this point in the revised manuscript.

b. Figure 1g. Please show more time points to demonstrate, that cKO-CT-CC Δ 73 (Tet-Off) /CENP-T Δ 161-200 and cKO-CT-CC Δ 73 (Tet-Off) /CENP-T Δ 201-216 are dead. The difference with one time point is not enough for this important conclusion.

We measured the cell numbers at all points, when the cells were viable. However, we could not plot when there were no viable cells. Therefore, we have added N.D. at the point where no viable cells were observed.

c. Figure 3c. Please provide more context for the observed increase of H3T3ph in cells with no Mis12. A brief discussion of this effect and references to prior publications would help to understand the importance of this observation.

We would like to emphasize that centromeric H3T3ph decreased and non-centromeric H3T3ph increased. We interpreted that the decrease in Mis12C caused a reduction in KNL1 and Bub1, which led to a reduction of H2A_T120ph and AuroraB at centromeres. H3T3ph could not be concentrated on such centromeres, and H3T3ph diffuses into the chromosome arm. We have explained this point in the revised manuscript.

d. Figure 3d. Selected images of cells do not correspond to quantifications. In WT cell image the level of Aurora B at the centromere is significantly higher than seen for CENP-T Δ 121-240 cell, whereas the quantification shows no statistical difference between these cells. The authors should provide more representative images.

We have changed representative images in the new Figure 3d, which correspond to quantification data in the revised version.

e. Figure S4b,c. Both panels show results for cKO-Nsl1 (AID) cells. In panel b level of GFP-AID-Nsl1 is auxin-dependent, however, analogous results in panel c show a lack

of auxin dependency. Authors should explain the design of this experiment (include diagram as in Figure S1) and why different results are seen in these two panels.

We apologize that the previous Figure S4b and c caused ambiguity. In cKO-Nsl1 cells, we did not detect a clear band for Nsl1-AID-GFP by immunoblotting. However, kinetochore signals for Nsl1-AID-GFP could be detected by microscopy. These kinetochore signals were not detected after IAA addition (Figure S4c). Figure S4d presents the expression of Nsl1-GFP (WT or EE mutant) on cKO-Nsl1 cells expressing additional Nsl1-GFP (WT or EE mutant) by immunoblot analysis. Therefore, Figure S4d does not present Nsl1-AID-GFP. We have explained this point clearly in the revised version. In addition, we have added a diagram for the creation of cKO-Nsl1 (AID) cells in revised Figure 4b.

*2) The authors could be more scholarly in their citation of the literature.
a) Some important statements in introduction are not supported by references. For example: "Furthermore, in addition to Knl1C-Mis12C-Ndc80C (KMN) formation, an additional copy of Ndc80C directly binds to CENP-T, and one KMN unit and one Ndc80C forms KMN-N in the network on CENP-T. The KMN network appears to play multiple roles, including in the binding of microtubules, spindle checkpoint functions, and cohesion of sister chromatids."*

We have added important references for these statements in the revised version.

b) Current paper shows that Ndc80 binding to CENP-T is required for Mis12 recruitment in cells. This has been shown previously (Hori et al., 2013; Rago et al, 2015). Proper discussion and references should be included to clarify which aspect of this result is considered novel by the authors, or this section should be toned down.

A previous conclusion that Ndc80 binding to CENP-T is required for Mis12 recruitment was reached from artificial tethering experiments using LacO-LacI. In this study, we presented this conclusion in native kinetochores. We have emphasized this point and described it clearly in the revised manuscript.

c) Page 5 and in discussion: "... the Ndc80C-Mis12C interaction is thought to be critical for the kinetochore functions..." Please improve citations. Alushin et al 2010 and DeLuca et al. 2006 do not have relevant data.

We have added the appropriate citations for this statement.

3) Legends to some figures do not explain which data are presented with "none" label. Please improve.

We have added the information for the "None" label in the revised version.

4) Page 12, line 9; page 13, lines 7,16; page 31 line 1. Authors claim that they perform "k-fiber microtubule staining". This is misleading, they perform staining of all spindle microtubules.

We have removed "k-fiber" from the revised version.

Reviewer #2

Faithful chromosome segregation is achieved thanks to the KMN network seeded on centromeres by two major pathways depending on either CENP-C or CENP-T. Their relative contributions to kinetochore function differ among species, and their redundancy is debated in the field. Here, Takenoshita et al tested the functional importance of distinct KMN sub-complexes recruited by CENP-T. Using the chicken DT40 cell system, they designed a comprehensive set of cell lines with various mutant

CENP-C and CENP-T alleles expressed under the endogenous promoters or under a Tet-Off promoter, which allowed them to dissect the contribution of each pathway to KMN network formation. They demonstrated that Kn11C and Mis12C (KM) components of the KMN network are functionally separate from Ndc80C (N). KM is necessary for localizing key mitotic kinases to centromeres and for centromere cohesion. Severing the KM-N interaction reduced Ndc80C levels at kinetochores and caused cell death, which was rescued by tethering additional copies of Ndc80C directly to CENP-T. The authors concluded that the Mis12C-Ndc80C interaction is not essential, and that KM and N components of KMN network are functionally distinct. Although the conclusions are largely consistent with previous work, the paper represents a significant addition to the kinetochore field as a thorough, precise, detailed analysis of the CENP-T pathway, taking advantage of state-of-the-art genetic complementation in the DT40 cell system. The data are high quality, and the results support the conclusions without requiring additional experiments, but the paper could benefit from some textual editing to improve clarity.

We thank you for appreciating our work. We have addressed the concerns you raised to improve the quality of this manuscript.

1. The number of repetitions performed for each experiment should be indicated throughout the paper. Fig 1e, for example, is presented as motivation for the later analyses, so it is important to know how much variation is observed in between repetitions.

We performed several repetitions for each experiment. We have described the number of repetitions for each experiment in the revised Figure legends.

2. Fig. 1d - The band marked by an asterisk could potentially be contamination with cells expressing 3xFLAG CENP-T WT, as the band migrates at exactly the same size. Can the authors use a different FLAG antibody exclude this possibility?

We tried to use different anti-FLAG antibodies (mouse or rat antibodies from different companies), albeit an asterisk band was always detected. We also used an anti-CENP-T antibody to detect 3xFLAG CENP-T. In this case, the asterisk band was not detected.

3. Fig. S1a – The authors should explain the reason for choosing the b-actin locus for expression of mScarlet-CENP-C .

To obtain cell lines constantly and stably expressing exogenous CENP-C, we chose the β -actin locus since β -actin is ubiquitously expressed. We have explained this point in the revised manuscript. Although we tried to introduce mScarlet-CENP-C downstream of the CENP-C promoter, it was difficult to do this owing to technical reasons; therefore, we have used the β -actin locus.

4. Fig. 1d and Fig. S2c – It looks like mScarlet-CENP-C migrates differently in cells without CENP-T expression or in cells expressing CENP-Tdelta90. Can the authors comment on this?

When cells were arrested at mitosis, CENP-C was highly phosphorylated, which led to a change in band mobility. We have described this point in the revised manuscript.

5. Fig 3e – It's difficult to see any difference in inter-kinetochore distance between the two images shown. It would help to measure the distances in these two examples and include that measurement in the figure to show that the examples are representative of the quantitation in the bar graph.

We repeated this experiment using an anti-FLAG antibody and have presented a better image (new Figure 3e), which represents data of the bar graph. Since

we measured the inter-kinetochore distance using 3D images, we could not include actual measurements in the 2D images.

6. Results (p 11) – “However, as in our previous model 37, a half reduction of Ndc80C levels on CENP-T was observed.” This sentence seems redundant with the previous sentence. If it is making an additional point, that should be explained more clearly.

We appreciate your comment and have revised the mentioned sentence carefully.

7. Results (p 14) – “Strikingly, expression of GFP-CENP-T2x(1-90) in cKO-Dsn1/Dsn1Δ326-349 cells completely compensated for growth defects in the cKO-Dsn1/Dsn1Δ326-349 cells in the presence of Tet (Figure 5g and S6d). These data indicate that Ndc80C does not necessarily bind to Mis12C.” If I understand the authors’ point here, the second sentence should be worded differently, such as “... does not necessarily need to bind ...” or the conclusion should be explained more clearly.

We appreciate this comment and have revised the mentioned sentence more clearly.

8. Fig. 6a – The CENP-C pathway should be included in the schematics because Dsn1 mutation affects CENP-C-KMN as well as CENP-T-KMN. Therefore, the phenotypes measured in this experiment will depend on changes in Ndc80 recruitment by both CENP-C and CENP-T pathways

We appreciate this comment and have added the schematic presentation for the CENP-C pathway in the revised Figure 6.

9. Discussion (p 16) – “Two copies of Ndc80C (N-N) are necessary and sufficient for proper kinetochore-microtubule attachment in DT40 cells, suggesting that binding of a single copy of Ndc80C to spindle microtubules is not enough to generate a force to pull a chromosome. Single-molecule experiments demonstrated that the multivalency of Ndc80C efficiently tracked depolymerizing microtubules, whereas a single copy of Ndc80C did not, which supports our observations.” This statement is misleading. Even with a single copy of Ndc80 per CENP-T, multivalent interactions could still occur due to multiple copies of CENP-T (and therefore multiple Ndc80 molecules) at each kinetochore. Can the authors speculate about why multiple copies of Ndc80 may need to bind to a single CENP-T?

We appreciate this comment and agree with this important point. As you suggested, if multiple copies of CENP-T localize to a kinetochore, the kinetochore would function. We believe that the number of CENP-T was limited, and this might be a reason why multiple copies of Ndc80C on a single CENP-T were necessary. We have added this point to the revised version.

Reviewer #3 (Ajit Joglekar)

The kinetochore is a complex organelle that is assembled in distinct protein layers to move chromosomes and to establish a platform for SAC signaling. The functional interface of the kinetochore consists of two critical protein components: the Ndc80 complex and the Knl1-Mis12 (KM) protein network. In some eukaryotes, this interface is assembled by two centromeric proteins: CENP-T and CENP-C. Using a series of elegant, cell biological experiments, Takenoshita and colleagues clarify the roles of Ndc80 and KM molecules that are recruited by CENP-C and CENP-T play in kinetochores in the DT40 chicken cells. They provide several insights into the assembly mechanisms of the DT40 kinetochores. (1) The recruitment of at least two Ndc80 molecules CENP-T is essential for the formation of proper kinetochore-microtubule attachment. (2) The two Ndc80 molecules per CENP-T can either be recruited via direct interaction with CENP-T or one via direct interaction and the other via Mis12 complex bound to CENP-T. (3) The recruitment and functions of Ndc80 and KM molecules recruited by CENP-T are separable in DT40 kinetochores.

These key findings clarify the inter-dependencies between the recruitment of Ndc80 complex and KM network molecules and show that their functions are separable. This is a key advance in our understanding of the eukaryotic kinetochore. The analysis presented by the authors is rigorous and extensive. For these reasons, I wholeheartedly support the publication of this manuscript with only minor changes to the manuscript. I do not think that any additional experiments are necessary.

We thank you for appreciating our work and have tried to incorporate all your constructive comments into the revised version.

1. In their cell growth assays, the authors describe reduced growth as a “growth delay” (e.g., at the top of page 7). This description isn’t quite accurate, because the retarded growth most likely results from increased cell death.

We appreciate this comment and have evaluated the cell viability. Since we did not find a significant increase in the number of dead cells, we believe that this phenotype showed a growth delay, possibly secondary to longer mitosis.

2. Page 8, 2nd para: “We examined other phosphorylation...” – these sentences need to be edited for clarity – e.g., “We examined the effects of CENP-T phosphorylation on Mis12 recruitment. There are eight phosphorylatable residues...”

We appreciate this comment. As per your suggestion, we have changed this sentence in the revised version.

3. Page 10, end of 2nd para: “we found that the distances between sister kinetochore pairs were increased” should be changed to: “distance between sister kinetochores increased”

We appreciate this comment. As per your suggestion, we have changed this sentence in the revised version.

4. Page 10, last para: “cohesion of sister chromatids or the recruitment of checkpoint proteins” should be changed to: cohesion of sister chromatids and the recruitment of checkpoint proteins

We appreciate this comment. As per your suggestion, we have changed this sentence in the revised version.

5. Page 11, 2nd para: “Although we found that one copy of Ndc80C still exists” should be changed to: “Although we found that one copy of Ndc80C per CENP-T still exists”

We appreciate this comment. As per your suggestion, we have changed this sentence in the revised version.

6. It will be useful to add key details and the citation for P the tension sensor probe in the main text, because there is also a FRET-based probe used to assess tension

We appreciate this comment. As per your suggestion, we have added a considerable amount of information for the tension sensor probe in the revised version.

7. Page 14, top: “These data indicate that Ndc80C does not necessarily bind to Mis12C” This is a very striking finding and needs to be explained more clearly. E.g., “Ndc80 need not interact with Mis12 for establishing MT-binding”. It might be a good idea to cite in vitro work from the Davis lab on yeast Ndc80-Mtw1 complex in the discussion, which suggests regulation of the MT-binding activity of Ndc80 through the Ndc80-Mtw1 interaction (PNAS).

We appreciate this comment. In the revised version, we have cited the

reference from the Davis lab, in which microtubule binding activity is regulated by Ndc80C-Mtw1 (Mis12C) interaction and discussed why direct interaction of Ndc80C with Mis12C is not necessary in vertebrates.

REVIEWERS' COMMENTS

Reviewer #1 (Remarks to the Author):

The authors responded to prior criticism in a very constructive and rigorous manner. They carried out additional experiments and significantly improved quantitative analysis, as well as provided additional controls. I fully support accepting this manuscript, however, I have one remaining concern that needs to be addressed.

1) Some of the newly added graphs lack statistical analysis: Fig 1j, Fig. 5 f and l, and Fig 6c. Please provide results of the tests comparing these distributions and the corresponding p-values. I do not believe that these p values will be fully consistent with the current statements in the text of the manuscript. Most notably the claim that two Ndc80s are fully sufficient to rescue the phenotype may not be supported by data in panel 5f . If this is the case, the authors statements should be modified accordingly.

Unfortunately, I do not think that the new title is an improvement. Previous title, in my opinion, was not very informative/novel, but the current title is just too cryptic. The authors should either provide a more clear title or perhaps revert to their original title. I will leave this decision to the authors. For what its worth, my own preference would be to use something along these lines: "Mis12-mediated recruitment of Ndc80 complex is not essential in human cells". Or "Recruitment of two Ndc80 complexes via CENP-T pathway is sufficient for cell viability". I realize that these titles do not cover all findings in this paper but I believe they are clear, informative and accurate.

Reviewer #2 (Remarks to the Author):

I recommend publication but suggest that the authors reconsider the wording of the title. The meaning of "dual Ndc80 complex" is unclear.

Reviewer #3 (Remarks to the Author):

The authors have satisfactorily addressed all the issues that I raised. The manuscript is much improved.

Response to Reviewers Takenoshita et al. (NCOMMS-21-26789A)

Reviewer #1

The authors responded to prior criticism in a very constructive and rigorous manner. They carried out additional experiments and significantly improved quantitative analysis, as well as provided additional controls. I fully support accepting this manuscript, however, I have one remaining concern that needs to be addressed.

1) Some of the newly added graphs lack statistical analysis: Fig 1j, Fig. 5 f and l, and Fig 6c. Please provide results of the tests comparing these distributions and the corresponding p-values. I do not believe that these p values will be fully consistent with the current statements in the text of the manuscript. Most notably the claim that two Ndc80s are fully sufficient to rescue the phenotype may not be supported by data in panel 5f. If this is the case, the authors statements should be modified accordingly.

We thank you for the appreciation of our work. According to your suggestions, we have added statistical analyses for Fig 1j, Fig. 5f and l, and Fig 6c. Then, based on data, we slightly modified description for Fig. 5f accordingly.

Unfortunately, I do not think that the new title is an improvement. Previous title, in my opinion, was not very informative/novel, but the current title is just too cryptic. The authors should either provide a more clear title or perhaps revert to their original title. I will leave this decision to the authors. For what its worth, my own preference would be to use something along these lines: "Mis12-mediated recruitment of Ndc80 complex is not essential in human cells". Or "Recruitment of two Ndc80 complexes via CENP-T pathway is sufficient for cell viability". I realize that these titles do not cover all findings in this paper but I believe they are clear, informative and accurate.

According to your suggestions, we have changed the title to "Recruitment of two Ndc80 complexes via the CENP-T pathway is sufficient for functional kinetochores" We believe that the new title reflects our main conclusions.

Reviewer #2

I recommend publication but suggest that the authors reconsider the wording of the title. The meaning of "dual Ndc80 complex" is unclear.

We thank you for appreciation of our work. As we mentioned above, we have changed to a new title in which we do not use "dual Ndc80 complex".

Reviewer #3

The authors have satisfactorily addressed all the issues that I raised. The manuscript is much improved.

We thank you for appreciation of our work.